# Modified Threshold Method for Ordinal Regression

## Abstract

Ordinal regression (OR, also called ordinal classification) is the classification of ordinal data in which the underlying target variable is discrete and has a natural ordinal relation. For OR problems, threshold methods are often employed since they are considered to capture the ordinal relation of data well: they learn a one-dimensional transformation (1DT) of an observation of the explanatory variables and classify the data by labeling that learned 1DT according to the rank of the interval to which the 1DT belongs among intervals on the real line separated by threshold parameters. In many conventional methods, the threshold parameters are determined regardless of the learning result of the 1DT and the task under consideration. Such conventional settings may deteriorate the classification performance. We, therefore, propose a novel computationally efficient method for determining the threshold parameters: it learns each threshold parameter independently through solving a problem relaxed from the minimization of the empirical task risk for the learned 1DT. The proposed labeling procedure experimentally gave superior classification performance with a feasible degree of additional computational load compared to four related existing labeling procedures in many of tried cases.

## 1 Introduction

***Ordinal regression (OR)*** (or called ordinal classification) is the classification of ***ordinal data*** in which the underlying target variable is discrete and labeled from a label set (ordinal scale) that is equipped with a natural ordinal relation; see Section 2.1 for a detailed formulation. The ordinal scale is typically formed as a graded (interval) summary of objective indicators like age groups {'0–9', '10–19', ..., '90–99', '100–'} or graded evaluation of subjectivity like human rating {'excellent', 'good', 'average', 'bad', 'terrible'}, and ordinal data appear in various practical applications: age estimation (Niu et al., 2016; Cao et al., 2020), information retrieval (Liu, 2011), movie rating (Yu et al., 2006), and questionnaire survey in social research (Bürkner & Vuorre, 2019).

***One-dimensional transformation (1DT)-based methods*** are often applied to the OR problems as a simple way to capture the ordinal relation of data. They typically learn a 1DT of an observation of the explanatory variables (and bias parameters of the number of classes minus one) and build a classifier based on that learned 1DT, as we will formalize in Section 2.2. Most existing 1DT-based methods are formulated in the form of employing one of four labeling procedures to map the learned 1DT to a class label. ***Threshold labelings*** used in ***threshold methods*** (or called threshold models) (Shashua & Levin, 2003; Chu & Keerthi, 2007; Pedregosa et al., 2017; Cao et al., 2020; Agarwal, 2008) include ***minimum threshold (MT)***, ***summation threshold (ST)***, and ***nearest-neighbor threshold (NNT) labelings***, which we will review in Sections 3.1, 3.2, and 3.3. They assign a label to the learned 1DT as the rank of the interval to which the 1DT belongs among intervals on the real line separated by threshold parameters. ***Likelihood-based (LB) labeling*** (see Section 3.4) is applied when the learning procedure uses a loss function characterizable as statistical modeling of conditional probabilities of the data and designed to minimize the task risk under the expectation that the model is correctly specified to the data distribution.

These existing labeling procedures, however, have respective concerns. The ST, MT, and NNT labelings use the learned bias parameters, their derivatives, or fixed points as the threshold parameters, but these settings are generally not motivated by minimizing the task risk. So their settings of the threshold parameters

actually can become sub-optimal for the minimization of the task risk, as we will demonstrate in Example 1. On the other hand, the LB labeling is designed to minimize the task risk if the assumed statistical model is correctly specified to the distribution of the data (see Theorem 1). However, its underlying statistical model has a strongly restricted representation ability owing to the use of the 1DT and can be misspecified to the data, so its performance can be degraded depending on the distribution of the data. In standard classification methods, a proper shape of the surrogate loss function guarantees the optimality of their decision boundaries irrelevant to the data distribution; recall Fisher consistency or classification calibration (Lin, 2004; Bartlett et al., 2006; Liu, 2007; Pires et al., 2013). In contrast, such a guarantee regarding the optimality of decision boundaries (threshold parameters if using a threshold labeling) does not hold in typical 1DT-based methods, as suggested also in (Pedregosa et al., 2017).

Previous studies have done little theoretical work on the properties of these existing labelings, so we first study the relationship between these labelings. In particular, we show in Theorem 2 that not only the MT, ST, and NNT labelings but also the LB labeling is a threshold labeling in typical usages. This finding motivates us to search for a better labeling function among the class of threshold labelings. Under the expectation that the 1DT is learned successfully and the empirical (training) task risk becomes a good estimator of the (test) task risk, we define the ***optimal threshold (OT) labeling*** as one that exactly minimizes the empirical task risk for the learned 1DT and propose to additionally learn the threshold parameters by minimizing that risk. A solution based on the brute-force search for the OT labeling is quite computationally demanding. Thus, we further propose a more computationally efficient alternative labeling, ***independently optimized threshold (IOT) labeling***: it applies threshold parameters each of which is independently learned through solving a problem relaxed from the minimization of the empirical task risk for the learned 1DT. Algorithm 1 for the IOT labeling can be performed with the computational complexity of quasi-linear order $O(n \log n)$ regarding the training sample size $n$. Also, as a performance guarantee of the IOT labeling, we show in Theorem 3 that the IOT labeling becomes the OT labeling as long as the resulting threshold parameters follow an appropriate order that we expected in formulating the relaxed problem to obtain threshold parameters for the IOT labeling.

In this study, we further took numerical experiments for the OR task for real-world ordinal data to confirm the practical effectiveness of the proposed IOT labeling (see Section 5). We then found following observations in many of tried cases:

- On the optimality condition (see Section 5.3.1): Algorithm 1 to determine the threshold parameters for the IOT labeling served appropriately ordered threshold parameters, which implies that a 1DT-based method with the IOT labeling gave a smaller training task risk than methods with other labeling functions for the same learned 1DT model.

- On the generalization performance (see Section 5.3.2): The IOT labeling gave superior generalization performance (more exactly, smaller test task risk) than the MT, ST, NNT, and LB labelings. Also, a modified threshold method with the IOT labeling outperformed an existing 1DT-based method using the ST labeling that has been declared by (Cao et al., 2020) to be state-of-the-art in 2020 for the age estimation from the facial image.

- On the computational efficiency (see Section 5.3.3): The IOT labeling is more disadvantageous in terms of the computational efficiency than the existing labelings as the training sample size $n$ increases, but Algorithm 1 for the IOT labeling just required additional computation time of fewer than 0.133 times the learning procedure for a DNN-based 1DT model even when $n \approx 118,000$. Namely, we confirmed that the IOT labeling is computationally feasible.

Therefore, this paper proposes a modified threshold method using the IOT labeling that could provide better classification performance than existing 1DT-based methods at the expense of some computational efficiency, on the ground of the fact (Theorem 2) that the MT, ST, and NNT labelings and LB labeling in typical usages are possibly sub-optimal threshold labelings, its conditional theoretical optimality (Theorem 3), and experimental effectiveness.

## 2 Preliminaries

### 2.1 Formulation of OR Problem

OR is the classification of ordinal data. The ordinal data have an underlying discrete **_target variable_** $Y \in [K] \coloneqq \{1, \ldots, K\}$ that is equipped with an ordinal relation naturally interpretable in the relationship with **_explanatory variables_** $\boldsymbol{X} \in \mathbb{R}^d$, where $K$ and $d$ are supposed to be integers larger than or equal to 3 and 1 respectively.[1] We here suppose that the target labels are encoded to $1, \ldots, K$ in an order-preserving manner, like from 'excellent', ..., 'terrible' to $1, \ldots, 5$.

The task of the OR is basically the same as that of the standard (including cost-sensitive) classification, to obtain a good **_classifier_** $f : \mathbb{R}^d \to [K]$. For a user-specified **_task loss_** $\ell : [K]^2 \to [0, \infty)$, it is formulated as minimization of the **_task risk_** $\mathbb{E}[\ell(f(\boldsymbol{X}), Y)]$, where the expectation value $\mathbb{E}[\cdot]$ is basically taken for all random variables in its argument (here $\boldsymbol{X}$ and $Y$). Popular task losses for OR tasks include not only the zero-one loss $\ell_{\mathrm{zo}}(j, k) \coloneqq \mathbf{1}_{j \neq k}$, where $\mathbf{1}_c$ takes 1 if a condition $c$ is true and 0 otherwise, but also V-shaped losses (for cost-sensitive tasks) reflecting one's preference of smaller prediction errors over larger ones such as the absolute deviation loss $\ell_{\mathrm{ad}}(j, k) \coloneqq |j - k|$, squared loss $\ell_{\mathrm{sq}}(j, k) \coloneqq (j - k)^2$, and $\ell_{\mathrm{zo},c}(j, k) \coloneqq \mathbf{1}_{|j-k|>c}$ with $c \geq 0$.

### 2.2 Formulation of 1DT-Based Methods and Threshold Methods

In this paper, we discuss only 1DT-based methods that have been developed in the OR research. We here provide notations and terminologies common for 1DT-based methods.

Many 1DT-based methods are designed according to the framework of surrogate risk minimization that allows for continuous optimization, so as to evade the difficulty of directly minimizing the task risk. They can have parameters $\boldsymbol{b} = (b_1, \ldots, b_{K-1}) \in \mathbb{R}^{K-1}$ (we call these the **_bias parameters_**) besides a **_1DT_** $g : \mathbb{R}^d \to \mathbb{R}$ to be learned respectively from the classes $\mathcal{B}$ and $\mathcal{G}$, and we denote a surrogate loss function used in such situations as $\phi(g(\boldsymbol{x}), \boldsymbol{b}, y)$ and call it the **_bias-parametric surrogate loss function_**. Note that the notations $b_0 \coloneqq -\infty$ and $b_K \coloneqq +\infty$ (and same ones added with bar or tilde symbol) are used together in the discussion for bias-parametric losses, to ease the description. On the other hand, a surrogate loss function used with no learnable bias parameters is called the **_bias-nonparametric surrogate loss function_** and denoted as $\phi(g(\boldsymbol{x}), y)$ with a 1DT $g \in \mathcal{G}$.

Suppose that one has the sample $\mathcal{D}_n \coloneqq \{(\boldsymbol{x}_i, y_i)\}_{i=1}^n$, each of which is drawn independently from an identical distribution of $(\boldsymbol{X}, Y)$. First, 1DT-based methods learn the 1DT model $g$ (and bias parameters $\boldsymbol{b}$) from the class $\mathcal{G}$ (and $\mathcal{B}$) through the minimization of the **_empirical surrogate risk_**, $\min_{g \in \mathcal{G}} \frac{1}{n} \sum_{i=1}^n \phi(g(\boldsymbol{x}_i), y_i)$ (or $\min_{g \in \mathcal{G}, \boldsymbol{b} \in \mathcal{B}} \frac{1}{n} \sum_{i=1}^n \phi(g(\boldsymbol{x}_i), \boldsymbol{b}, y_i)$), or its regularized version. For the class $\mathcal{B}$ when using a bias-parametric loss, several methods impose the ordering constraint $b_1 \leq \cdots \leq b_{K-1}$ on the bias parameters $\boldsymbol{b}$. As one way to implement this constraint, Franses & Paap (2001) mentioned to parametrize the bias parameters $\boldsymbol{b}$ as

$$b_1 = b_1', \text{ and } b_k = b_{k-1} + b_k'^2 \text{ for } k = 2, \ldots, K-1, \tag{1}$$

with other parameters $b_1', \ldots, b_{K-1}' \in \mathbb{R}$.

Next, 1DT-based methods build up a classifier as $f = h \circ \bar{g}$ with a learned 1DT $\bar{g}$ and a labeling function $h : \mathbb{R} \to [K]$. Most existing 1DT-based methods can be seen as adopting one of the MT, ST, NNT, and LB labelings, depending on the properties of their surrogate loss like being bias-parametric or bias-nonparametric, and statistical or non-statistical (defined in Section 3.4), as we will review in the succeeding

---

[1] For better modeling of the ordinal data, it would be important to provide a mathematical characterization and further discussion of the natural ordinal relation. However, it would have a relation closer to the learning procedure of the 1DT and bias parameters (defined in Section 2.2), and its necessity is not so great for the analysis and proposal of this study, so we will not mention it in this paper. Refer to, for example, an OR study (da Costa et al., 2008) for the discussion on such characterizations.

[2] Simplification of description is applied to this formulation: Let $\sum_{k=l}^m f(k)$ be 0 irrelevant to the function $f$ if $l > m$, and $\varphi(g(\boldsymbol{x}) - b_0) = \varphi(b_K - g(\boldsymbol{x})) = 0$.

Table 1: It shows classes of surrogate loss functions used for 1DT-based methods, their representative instances, and commonly-used labelings. Note that we found no widely-used bias-nonparametric statistical losses and that our proposed IOT labeling is applicable along with every loss function.[2]

| Classes | Surrogate Loss Functions
Representative Instances
Commonly-Used Labelings |
|---|---|
| Non-statistical
(bias-parametric) | Immediate threshold (IT) $\phi(g(\boldsymbol{x}), \boldsymbol{b}, y) = \varphi(g(\boldsymbol{x}) - b_{y-1}) + \varphi(b_y - g(\boldsymbol{x}))$,
All threshold (AT) $\phi(g(\boldsymbol{x}), \boldsymbol{b}, y) = \sum_{k=1}^{y-1} \varphi(g(\boldsymbol{x}) - b_k) + \sum_{k=y}^{K-1} \varphi(b_k - g(\boldsymbol{x}))$
$\varphi$ is popular in binary classification: $\varphi(u) = \max\{1-u, 0\}$
(Shashua & Levin, 2003), $e^{-u}$ (Lin & Li, 2006)
MT (Shashua & Levin, 2003; Chu & Keerthi, 2007), ST (Pedregosa et al., 2017) |
| Statistical
(bias-parametric) | NLL $\phi(g(\boldsymbol{x}), \boldsymbol{b}, y) = -\log(\sigma(b_y - g(\boldsymbol{x})) - \sigma(b_{y-1} - g(\boldsymbol{x})))$,
AT with $\varphi(u) = -\log(\sigma(u))$ (ANLCL)
$\sigma$ is a CDF: $\sigma(u) = 1/(1 + e^{-u})$ (McCullagh, 1980),
$\int_{-\infty}^{u} (2\pi)^{-1/2} e^{-v^2/2}\, dv$ (Chu & Ghahramani, 2005)
LB (McCullagh, 1980), ST (Cao et al., 2020) |
| Bias-nonparametric
(non-statistical) | Regression $\phi(g(\boldsymbol{x}), y) = \varphi(y - g(\boldsymbol{x}))$
$\varphi$ is popular in regression: $\varphi(u) = |u|$ (Agarwal, 2008), $u^2$ (Kramer et al., 2001)
NNT (Agarwal, 2008) |

sections; see also Table 1. In particular, a 1DT-based method that uses a ***threshold labeling***

$$h_{\mathrm{thr}}(u; \boldsymbol{t}) := 1 + \sum_{k=1}^{K-1} \mathbf{1}_{u \geq t_k} \tag{2}$$

as its labeling function $h$ is called the ***threshold method***, where we call $\boldsymbol{t} = (t_1, \ldots, t_{K-1}) \in \mathbb{R}^{K-1}$ the ***threshold parameters***.[3] Note that the threshold labeling $h_{\mathrm{thr}}(u; \boldsymbol{t})$ has the following properties:

**Proposition 1.** *The threshold labeling $h_{\mathrm{thr}}(u; \boldsymbol{t})$ is non-decreasing and right-continuous in $u \in \mathbb{R}$ and invariant regarding the permutation of the threshold parameters $t_1, \ldots, t_{K-1}$. Conversely, an arbitrary non-decreasing right-continuous function $h : \mathbb{R} \to [K]$ can be represented by a threshold labeling $h_{\mathrm{thr}}(\cdot; \boldsymbol{t})$ with certain threshold parameters $\boldsymbol{t} \in \mathbb{R}^{K-1}$ (i.e., there exist $\boldsymbol{t} \in \mathbb{R}^{K-1}$ such that $h(u) = h_{\mathrm{thr}}(u; \boldsymbol{t})$ for any $u \in \mathbb{R}$) or their permutation. Also, if $t_1, \ldots, t_{K-1}$ take only $L$ different values, then $h_{\mathrm{thr}}(u; \boldsymbol{t})$ has $L$ change points $u = u_1, \ldots, u_L$ such that $h_{\mathrm{thr}}(u_l - \epsilon; \boldsymbol{t}) \neq h_{\mathrm{thr}}(u_l; \boldsymbol{t})$ with a sufficiently small $\epsilon > 0$ for $l = 1, \ldots, L$.*

The last result implies that the threshold labeling is the simplest as the labeling function in the sense that the resulting classifier has only $(K-1)$ decision boundaries for the learned 1DT at most.

## 2.3 Formulation of Policy of This Study

This study aims for a better labeling, especially for a better threshold labeling. Thus, regarding the learning procedure of a 1DT (and bias parameters), that is, the surrogate loss $\phi$ and class $\mathcal{G}$ (and $\mathcal{B}$), this paper adopts those by existing studies, and we do not discuss the goodness of the learning procedure. Assuming that a learned 1DT $\bar{g}$ is given, we will discuss the goodness of the labeling function $h$ with the task risk $\mathbb{E}[\ell(h(\bar{g}(\boldsymbol{X})), Y)]$ or the ***empirical task risk*** $\frac{1}{n} \sum_{i=1}^{n} \ell(h(\bar{g}(\boldsymbol{x}_i)), y_i)$ as the criterion.

---

[3]As we will review in Section 3.2, the ST labeling is a threshold labeling that applies the learned bias parameters as the threshold parameters. Therefore, several studies on the threshold methods supposing to use the ST labeling formulate the methods without distinguishing the bias and threshold parameters, unlike our formulation; ours can be seen as a generalization of such previous one and is also a contribution of this paper.

# 3 Review and Analysis of Existing 1DT-Based Methods and Labeling Functions

## 3.1 Minimum Threshold (MT) Labeling

Threshold methods have been studied actively in the machine learning literature. Most of early threshold methods using a bias-parametric surrogate loss have been formulated with the MT labeling.

The previous OR studies (Herbrich, 2000; Shashua & Levin, 2003; Lin & Li, 2006; Chu & Keerthi, 2007) have developed various large-margin-type methods. For instance, ***support vector OR (SVOR)*** proposed by Shashua & Levin (2003) learns the 1DT $g$ and bias parameters $\boldsymbol{b}$ for the data point $(\boldsymbol{X}, Y) = (\boldsymbol{x}, y)$ with the bias-parametric surrogate loss

$$\phi_{\mathrm{svor}}(g(\boldsymbol{x}), \boldsymbol{b}, y) := \max\{1 + b_{y-1} - g(\boldsymbol{x}), 0\} + \max\{1 - b_y + g(\boldsymbol{x}), 0\}, \tag{3}$$

under the class restriction $\mathcal{G} = \{g : \mathbb{R}^d \to \mathbb{R}\}$ and $\mathcal{B} = \mathbb{R}^{K-1}$. Chu & Keerthi (2007) proposed to impose the explicit ordering constraint on the bias parameters during the learning procedure, that is, it instead uses the class $\mathcal{B} = \{\boldsymbol{b} \in \mathbb{R}^{K-1} \mid b_1 \leq \cdots \leq b_{K-1}\}$. They have proposed the methods, on the basis of the MT labeling $h_{\mathrm{mt}}(\cdot; \bar{\boldsymbol{b}})$ with the learned bias parameters $\bar{\boldsymbol{b}}$, namely their methods predict a label of $\boldsymbol{X} = \boldsymbol{x}$ as

$$f(\boldsymbol{x}) = h_{\mathrm{mt}}(\bar{g}(\boldsymbol{x}); \bar{\boldsymbol{b}}), \text{ with } h_{\mathrm{mt}}(u; \boldsymbol{b}) := \min\{k \in [K] \mid u < b_k\}. \tag{4}$$

The MT labeling is a threshold labeling and has the following relationship to the formulation of the threshold labeling (2):

**Proposition 2.** *Given $\bar{\boldsymbol{b}} \in \mathbb{R}^{K-1}$ together with $\bar{b}_0 := -\infty$ and $\bar{b}_K := +\infty$, let $t_k$ be $\bar{b}_{i_k}$ with $i_k := \min\{j \in \{0, \ldots, k\} \mid \bar{b}_k \leq \bar{b}_j\}$ for each $k = 1, \ldots, K-1$. Then, one has that $h_{\mathrm{mt}}(u; \bar{\boldsymbol{b}}) = h_{\mathrm{thr}}(u; \boldsymbol{t})$ with $\boldsymbol{t} = (t_1, \ldots, t_{K-1})$. Also, $h_{\mathrm{mt}}(u; \bar{\boldsymbol{b}}) = h_{\mathrm{thr}}(u; \bar{\boldsymbol{b}})$ if $\bar{b}_1 \leq \cdots \leq \bar{b}_{K-1}$.*

Using the MT labeling $h_{\mathrm{mt}}(\cdot; \bar{\boldsymbol{b}})$ implies that decision boundaries are set depending only on the values of learned bias parameters $\bar{\boldsymbol{b}}$. This setting is almost like a convention, and its goodness is not supported by theoretical discussion. This convention may be a negative factor that degrades the classification performance of threshold methods, as demonstrated below:

**Example 1.** *The surrogate loss function $\phi_{\mathrm{svor}}(\cdot, \boldsymbol{b}, k)$ of SVOR is a continuous convex upper bound of the zero-one task loss function with the MT labeling $\ell_{\mathrm{zo}}(h_{\mathrm{mt}}(\cdot; \boldsymbol{b}), k)$ when $\boldsymbol{b}$ is appropriately ordered (i.e., $b_1 \leq \cdots \leq b_{K-1}$): $\phi_{\mathrm{svor}}(\cdot, \boldsymbol{b}, k)$ is a convex function and $\phi_{\mathrm{svor}}(\cdot, \boldsymbol{b}, k) \geq \ell_{\mathrm{zo}}(h_{\mathrm{mt}}(\cdot; \boldsymbol{b}), k)$. So one may think that SVOR and the task with the zero-one task loss (**Task-Z**) have friendly compatibility, from an analogy of a well-known result, classification calibration (Bartlett et al., 2006), in binary classification. However, the following demonstration shows that the MT labeling may be sub-optimal in minimizing the task risk as a labeling function for the combination of SVOR and Task-Z.*

*We consider a 4-class OR problem (let $K = 4$), and suppose that the data appear only on 4 different points $\boldsymbol{x}^{[1]}, \ldots, \boldsymbol{x}^{[4]}$ in $\mathbb{R}^d$ and follow the probability distribution, $\Pr(\boldsymbol{x}^{[i]}) = 0.25$ and $(\Pr(1|\boldsymbol{x}^{[i]}), \ldots, \Pr(K|\boldsymbol{x}^{[i]})) = (0.5, 0.4, 0, 0.1), (0.3, 0.5, 0, 0.2), (0.2, 0, 0.5, 0.3), (0.1, 0, 0.4, 0.5)$ for $i = 1, \ldots, 4$.[4]*

*It can be shown that the surrogate risk minimizer $\{\bar{g}, \bar{\boldsymbol{b}}\} := \arg\min_{g \in \mathcal{G}, \boldsymbol{b} \in \mathcal{B}} \mathbb{E}[\phi_{\mathrm{svor}}(g(\boldsymbol{X}), \boldsymbol{b}, Y)]$ satisfies $\bar{b}_1 = \bar{b}_2 = \bar{b}_3 = 0$, $\bar{g}(\boldsymbol{x}^{[1]}) = \bar{g}(\boldsymbol{x}^{[2]}) = -1$, and $\bar{g}(\boldsymbol{x}^{[3]}) = \bar{g}(\boldsymbol{x}^{[4]}) = 1$ (ignore the translation invariance) by several simple calculations. The MT labeling predicts a label of the data on $\boldsymbol{x}^{[i]}$ as $h_{\mathrm{mt}}(\bar{g}(\boldsymbol{x}^{[i]}); \bar{\boldsymbol{b}}) = 1, 1, 4, 4$ for $i = 1, \ldots, 4$ ($\mathbb{E}[\ell_{\mathrm{zo}}(h_{\mathrm{mt}}(\bar{g}(\boldsymbol{X}); \bar{\boldsymbol{b}}), Y)] = 0.6$), despite that using different threshold parameters (say $\boldsymbol{t} = (-2, 0, 2)$) can predict it as $h_{\mathrm{mt}}(\bar{g}(\boldsymbol{x}^{[i]}); \boldsymbol{t}) = 2, 2, 3, 3$ for $i = 1, \ldots, 4$ and yield a smaller value of the task risk ($\mathbb{E}[\ell_{\mathrm{zo}}(h_{\mathrm{mt}}(\bar{g}(\boldsymbol{X}); \boldsymbol{t}), Y)] = 0.55$).* □

## 3.2 Summation Threshold (ST) Labeling

More recent papers in machine learning (Pedregosa et al., 2017; Cao et al., 2020) have discussed and proposed threshold methods that are similar to ones reviewed in the previous section but formulated based on the

---

[4]We abbreviate the marginal probability $\Pr(\boldsymbol{X} = \boldsymbol{x})$ to $\Pr(\boldsymbol{x})$ and the conditional probability $\Pr(Y = y | \boldsymbol{X} = \boldsymbol{x})$ to $\Pr(y|\boldsymbol{x})$ (this abbreviation applies to an estimate $\hat{\Pr}$ as well).

different ST labeling. The ST labeling is also a certain threshold labeling $h_{\text{thr}}(\cdot; \boldsymbol{t})$ with threshold parameters $\boldsymbol{t}$ depending only on the values of learned bias parameters $\bar{\boldsymbol{b}}$, and the classifier can be represented in the notation of this paper as

$$f(\boldsymbol{x}) = h_{\text{st}}(\bar{g}(\boldsymbol{x}); \bar{\boldsymbol{b}}), \text{ with } h_{\text{st}}(u; \boldsymbol{b}) = h_{\text{thr}}(u; \boldsymbol{b}). \tag{5}$$

While the difference between the MT labeling $h_{\text{mt}}(u; \bar{\boldsymbol{b}})$ and ST labeling $h_{\text{st}}(u; \bar{\boldsymbol{b}})$ and the significance of the difference are discussed little in the existing research, we have to remark that they are different when the learned bias parameters $\bar{\boldsymbol{b}}$ are not ordered; recall Proposition 2.

What is important in our discussion is that the setting $\boldsymbol{t} = \bar{\boldsymbol{b}}$ for threshold labeling $h_{\text{thr}}(\cdot; \boldsymbol{t})$ is not motivated by minimization of the task risk and may degrade the classification performance of threshold methods, as in the case of the MT labeling. Threshold parameters $\bar{\boldsymbol{b}}$ of the ST labeling are also sub-optimal as the threshold parameters under the setting in Example 1, since $h_{\text{mt}}(\cdot; \bar{\boldsymbol{b}}) = h_{\text{st}}(\cdot; \bar{\boldsymbol{b}})$ there.

### 3.3 Nearest-Neighbor Threshold (NNT) Labeling

Bias-nonparametric surrogate losses are often applied for OR problems together with the NNT labeling, a threshold labeling that rounds the learned 1DT to its nearest label.

For instance, Agarwal (2008) used the ***absolute-deviation (AD) loss*** $\phi_{\text{ad}}(g(\boldsymbol{x}), y) := |y - g(\boldsymbol{x})|$ for the data point $(\boldsymbol{x}, y)$ to learn a 1DT model $g$, and makes a label prediction via the threshold labeling $h_{\text{thr}}(\cdot; \boldsymbol{t})$ with the threshold parameters $t_k = k + 1/2$, $k = 1, \ldots, K - 1$.

Similarly to the MT and ST labelings used with bias-parametric losses, the threshold parameters of the NNT labeling may not be optimal for minimizing the task risk depending on the underlying data distribution and task loss, since threshold parameters are determined without considering both the learned 1DT and the task to solve.

### 3.4 Likelihood-Based (LB) Labeling

In the OR research based on statistics, several methods have been developed according to the statistical modeling of the conditional probabilities of the data through a 1DT (McCullagh, 1980; Williams, 2006; Chu & Ghahramani, 2005). They apply bias-parametric statistical surrogate loss functions associated with their statistical modeling, where we call a loss function designed based on the modeling of conditional probabilities of data as the ***statistical surrogate loss function***. For such methods, not only the threshold labelings but also the LB labeling that grounds on their assumed statistical model is a commonly-used option for the labeling function.

For example, ***ordinal logistic regression (OLR)*** (McCullagh, 1980) models the conditional probabilities $\Pr(y|\boldsymbol{x})$, $(\boldsymbol{x}, y) \in \mathbb{R}^d \times [K]$ by

$$\hat{\Pr}(y|\boldsymbol{x}; \tilde{g}, \tilde{\boldsymbol{b}}) := \sigma(\tilde{b}_y - \tilde{g}(\boldsymbol{x})) - \sigma(\tilde{b}_{y-1} - \tilde{g}(\boldsymbol{x})), \tag{6}$$

with the sigmoid function $\sigma = \sigma_{\text{olr}}$ with $\sigma_{\text{olr}}(u) := 1/(1 + e^{-u})$, assumed 1DT $\tilde{g} : \mathbb{R}^d \to \mathbb{R}$, assumed bias parameters $\tilde{\boldsymbol{b}} := (\tilde{b}_1, \ldots, \tilde{b}_{K-1}) \in \mathbb{R}^{K-1}$, $\tilde{b}_0 := -\infty$, and $\tilde{b}_K := +\infty$. Other options for the link function $\sigma : \mathbb{R} \to [0, 1]$ should satisfy the properties of a cumulative distribution function (CDF) that is non-decreasing and $\sigma(-\infty) = 0$ and $\sigma(+\infty) = 1$, so that the model (6) is normalized, i.e., $\sum_{y=1}^{K} \hat{\Pr}(y|\boldsymbol{x}; \tilde{g}, \tilde{\boldsymbol{b}}) = 1$. Gaussian process OR (GPOR) proposed by Chu & Ghahramani (2005) uses the CPD function of the standard Gaussian distribution (a.k.a. the inverse function of the probit function) $\sigma_{\text{gpor}}(u) := \int_{-\infty}^{u} (2\pi)^{-1/2} e^{-v^2/2} \, dv$. Also, they assume that bias parameters $\tilde{b}_k$, $k = 1, \ldots, K - 1$ are non-decreasing in the index so that the conditional probability model $\hat{\Pr}(y|\boldsymbol{x}; \tilde{g}; \tilde{\boldsymbol{b}})$ gets non-negative for any $(\boldsymbol{x}, y) \in \mathbb{R}^d \times [K]$ and $\tilde{g}$ under the non-decreasingness of $\sigma$.

These methods typically learn the 1DT model $g$ and bias parameters $\boldsymbol{b}$ for the data $(\boldsymbol{X}, Y) = (\boldsymbol{x}, y)$ with the bias-parametric surrogate loss function

$$\phi_{\text{nll}}(g(\boldsymbol{x}), \boldsymbol{b}, y) := -\log(\sigma(b_y - g(\boldsymbol{x})) - \sigma(b_{y-1} - g(\boldsymbol{x}))), \tag{7}$$

which amounts to the ***negative log likelihood (NLL) loss function*** for the specified statistical model (6). Also, Cao et al. (2020) used (for $\sigma = \sigma_{\mathrm{olr}}$) another loss function

$$\phi_{\mathrm{anlcl}}(g(\boldsymbol{x}), \boldsymbol{b}, y) := -\sum_{k=1}^{y-1} \log(\sigma(g(\boldsymbol{x}) - b_k)) - \sum_{k=y}^{K-1} \log(\sigma(b_k - g(\boldsymbol{x}))), \tag{8}$$

which we call the ***all negative log cumulative likelihoods (ANLCL) loss function***. The learning procedure for this loss function is characterized as the minimization of sum of the NLLs of the models of cumulative conditional probabilities $\Pr(Y \le k | \boldsymbol{X} = \boldsymbol{x})$ for binary classification problems, '$k$ or less' vs. 'more than $k$', $k = 1, \ldots, K-1$.

The above interpretation on using the surrogate losses $\phi_{\mathrm{nll}}$ and $\phi_{\mathrm{anlcl}}$ under the statistical model (6) can be mathematically understood as follows:

**Theorem 1.** *Assume that the random variable $(\boldsymbol{X}, Y)$ underlying the data has conditional probabilities that can be represented as (6): $\Pr(y|\boldsymbol{x}) = \hat{\Pr}(y|\boldsymbol{x}; \tilde{g}, \tilde{\boldsymbol{b}})$ for every $y \in [K]$ and any $\boldsymbol{x} \in \mathbb{R}^d$ in the support of the distribution of $\boldsymbol{X}$ with $\sigma$ that is non-decreasing and satisfies $\sigma(-\infty) = 0$ and $\sigma(+\infty) = 1$ (and $\sigma(-\cdot) = 1 - \sigma(\cdot)$ for $\phi = \phi_{\mathrm{anlcl}}$) such as $\sigma_{\mathrm{olr}}$ and $\sigma_{\mathrm{gpor}}$, $\tilde{g} : \mathbb{R}^d \to \mathbb{R}$, and $\tilde{\boldsymbol{b}} \in \mathbb{R}^{K-1}$ satisfying $\tilde{b}_1 \le \cdots \le \tilde{b}_{K-1}$. Let $(\phi, \mathcal{B})$ be $(\phi_{\mathrm{nll}}, \{\boldsymbol{b} \in \mathbb{R}^{K-1} \mid b_1 \le \cdots \le b_{K-1}\})$, $(\phi_{\mathrm{anlcl}}, \mathbb{R}^{K-1})$, or $(\phi_{\mathrm{anlcl}}, \{\boldsymbol{b} \in \mathbb{R}^{K-1} \mid b_1 \le \cdots \le b_{K-1}\})$, and $\mathcal{G}$ be $\{g : \mathbb{R}^d \to \mathbb{R}\}$. Then, any surrogate risk minimizer $\{\bar{g}, \bar{\boldsymbol{b}}\} \in \arg\min_{g \in \mathcal{G}, \boldsymbol{b} \in \mathcal{B}} \mathbb{E}[\phi(g(\boldsymbol{X}), \boldsymbol{b}, Y)]$ satisfies $\hat{\Pr}(y|\boldsymbol{x}; \bar{g}, \bar{\boldsymbol{b}}) = \Pr(y|\boldsymbol{x})$ for any $\boldsymbol{x} \in \mathbb{R}^d$ in the support of the distribution of $\boldsymbol{X}$.*

Note that such a characterization has not been known for the (non-statistical) surrogate loss of SVOR reviewed in the previous section.

Considering Theorem 1 and the equality $\mathbb{E}[\ell(f(\boldsymbol{x}), Y)] = \sum_{k=1}^{K} \Pr(k|\boldsymbol{x})\ell(f(\boldsymbol{x}), k)$, and aiming to minimize the task risk, $\min_{f:\mathbb{R}^d \to [K]} \mathbb{E}[\ell(f(X), Y)]$, these methods can predict a label of an observation $\boldsymbol{X} = \boldsymbol{x}$ by the classifier

$$f(\boldsymbol{x}) = h_{\mathrm{lb}}(\bar{g}(\boldsymbol{x}); \bar{\boldsymbol{b}}, \ell) = \arg\min_{j \in [K]} \sum_{k=1}^{K} \hat{\Pr}(k|\boldsymbol{x}; \bar{g}, \bar{\boldsymbol{b}})\ell(j, k) \tag{9}$$

with learned 1DT $\bar{g}$, learned bias parameters $\bar{\boldsymbol{b}}$, and the LB labeling

$$h_{\mathrm{lb}}(u; \boldsymbol{b}, \ell) := \arg\min_{j \in [K]} \sum_{k=1}^{K} \{\sigma(b_k - u) - \sigma(b_{k-1} - u)\}\ell(j, k), \tag{10}$$

under the expectation that the assumed statistical model (6) correctly represents the actual statistical behavior of the data and it is learned successfully.[5]

These methods tend to perform better when their assumed statistical model represents the actual statistical behavior of the data well. One can, however, find that the condition in Theorem 1 is very restrictive. Therefore, in many practical situations, their statistical model would deviate from the actual statistical behavior of the data, and then their 1DT model may not be learned appropriately, and the LB labeling $h_{\mathrm{lb}}(\cdot; \bar{\boldsymbol{b}}, \ell)$ may be sub-optimal for the learned 1DT model $\bar{g}$.

One may still consider that the LB labeling is more flexible, in that it is generally not restricted within the class of non-decreasing threshold labelings, and superior to threshold labelings. However, we found that the LB labeling takes the form of the threshold labeling, for typical statistical models such as ones in OLR and GPOR (i.e., the link function $\sigma$ such as $\sigma_{\mathrm{olr}}, \sigma_{\mathrm{gpor}}$) and for typical task losses such as $\ell = \ell_{\mathrm{zo}}, \ell_{\mathrm{zo},c}, \ell_{\mathrm{ad}}, \ell_{\mathrm{sq}}$.

**Theorem 2.** *Suppose that $\sigma$ is non-decreasing and satisfies $\sigma(-\infty) = 0$ and $\sigma(+\infty) = 1$ and that $\bar{b}_1 \le \cdots \le \bar{b}_{K-1}$. Then, the LB labeling $h_{\mathrm{lb}}(u; \bar{\boldsymbol{b}}, \ell)$ is*

(i) *a certain threshold labeling $h_{\mathrm{thr}}(u; \boldsymbol{t})$ for some $\boldsymbol{t} \in \mathbb{R}^{K-1}$, if $\ell(k, l)$ at each fixed $k \in [K]$ is non-increasing in $l$ for $l \le k$ and non-decreasing in $l$ for $l \ge k$, and $\ell_{k,l}(j) := \ell(k, j) - \ell(k, j+1) - \ell(l, j) +$*

---

[5]There can be a situation where objective functions with different $j$ of (10) take the same value in principle, but such a situation hardly occurs. Thus, this paper assumes that such a situation does not occur at all and does not discuss it.

$\ell(l, j + 1)$ *at each fixed different* $k, l \in [K]$ *is non-positive (resp. non-negative) for all* $j \in [K - 1]$ *respectively when* $k < l$ *(resp.* $k > l$*), such as* $\ell = \ell_{\mathrm{ad}}, \ell_{\mathrm{sq}}$,

(ii) *a certain threshold labeling* $h_{\mathrm{thr}}(u; \boldsymbol{t})$ *for some* $\boldsymbol{t} \in \mathbb{R}^{K-1}$, *if* $\ell = \ell_{\mathrm{zo}}, \ell_{\mathrm{zo}, c}$ *with* $c \in [0, \lfloor K/2 \rfloor)$, $\sigma$ *is differentiable,* $\sigma'(v)$ *is even and non-increasing in* $v$ *if* $v > 0$, *and* $\frac{\sigma'(v_1) - \sigma'(v_2)}{\sigma(v_1) - \sigma(v_2)}$ *is non-increasing in* $v_1$ *with fixed* $v_2$ *and in* $v_2$ *with fixed* $v_1$ *if* $v_1 < v_2$, *such as* $\sigma_{\mathrm{olr}}$ *and* $\sigma_{\mathrm{gpor}}$, *where* $\lfloor v \rfloor$ *is the greatest integer less than or equal to* $v$,

(iii) *the threshold labeling* $h_{\mathrm{thr}}(u; \bar{\boldsymbol{b}})$ *that is same as the MT labeling* $h_{\mathrm{mt}}(u; \bar{\boldsymbol{b}})$ *and ST labeling* $h_{\mathrm{st}}(u; \bar{\boldsymbol{b}})$, *if* $\ell = \ell_{\mathrm{ad}}$ *and* $\sigma(0) = 0.5$.

Here, Theorem 2 (i) assumes that the task loss $\ell$ is V-shaped, and the condition on $\ell_{k,l}$ in Theorem 2 (i) holds under the convexity of $\ell$ defined below:

**Corollary 1.** $\ell_{k,l}(j)$ *at each fixed different* $k, l \in [K]$ *is non-positive and non-negative for all* $j \in [K - 1]$ *respectively when* $k < l$ *and* $k > l$, *if the task risk* $\ell$ *is convex in the difference of the two arguments:*

$$\ell(j_3, k_3) \le \frac{(j_3 - k_3) - (j_1 - k_1)}{(j_2 - k_2) - (j_1 - k_1)} \ell(j_1, k_1) + \frac{(j_2 - k_2) - (j_3 - k_3)}{(j_2 - k_2) - (j_1 - k_1)} \ell(j_2, k_2) \tag{11}$$

*for all* $j_1, \ldots, k_3 \in [K]$ *such that* $j_1 - k_1 \ne j_2 - k_2$ *and* $j_1 - k_1 \le j_3 - k_3 \le j_2 - k_2$.

The condition on $\sigma$ in Theorem 2 (ii) comes from the consideration for non-convex task losses.

## 4 Our Proposal: Independently Optimized Threshold (IOT) Labeling

### 4.1 Optimal Threshold (OT) Labeling

In typical usages, not only the MT, ST, and NNT labelings, but also the LB labeling is a threshold labeling, as we confirmed in Theorem 2. Thus, we consider that it would be meaningful to aim for a better threshold labeling for improving the classification performance of existing 1DT-based methods. Recalling that the final goal is to make the task risk $\mathbb{E}[\ell(f(\boldsymbol{X}), Y)]$ small, and expecting that the 1DT was learned successfully and the empirical (training) task risk becomes a good estimator of the (test) task risk, we adopt the empirical task risk,

$$R(\boldsymbol{t}; \ell, \bar{g}, \mathcal{D}_n) := \frac{1}{n} \sum_{i=1}^{n} \ell(h_{\mathrm{thr}}(\bar{g}(\boldsymbol{x}_i); \boldsymbol{t}), y_i) \tag{12}$$

for a given learned 1DT model $\bar{g}$, as the optimality criterion for the threshold parameters:

**Definition 1.** *We call the threshold parameters* $\bar{\boldsymbol{t}}_{\mathrm{ot}} \in \arg\min_{\boldsymbol{t} \in \mathbb{R}^{K-1}} R(\boldsymbol{t}; \ell, \bar{g}, \mathcal{D}_n)$ *as the* ***OT parameters*** *and the corresponding threshold labeling* $h_{\mathrm{thr}}(\cdot; \bar{\boldsymbol{t}}_{\mathrm{ot}})$ *as the* ***OT labeling***.

Accordingly, we further propose to learn the threshold parameters $\boldsymbol{t}$ by minimizing the objective function $R(\boldsymbol{t}; \ell, \bar{g}, \mathcal{D}_n)$.

We have to provide two remarks on the additional learning of the decision boundaries (here threshold parameters) after the learning of the learner model (here a 1DT). The first remark is that, although the additional learning can be applied to other classification methods such as binary classification methods, its significance for 1DT-based methods stems from the fact that the uniformly (namely, irrelevant to the data distribution) optimal decision boundaries are not known for 1DT-based methods in many cases. For example, Lin (2004); Bartlett et al. (2006); Liu (2007); Pires et al. (2013) showed that well-known methods in standard (including multi-class and cost-sensitive) classification have such an optimality guarantee, but most 1DT-based methods do not, as demonstrated below:

**Example 2.** *This example shows that a threshold labeling for SVOR does not have a guarantee of the optimality in the sense of the Bayes optimality.*

*Assume the setting in Example 1. The Bayes optimal classifier $\bar{f} := \arg\min_{f:\mathbb{R}^d \to [K]} \mathbb{E}[\ell_{\mathrm{zo}}(f(\boldsymbol{X}), Y)]$ predicts a label of the data on $\boldsymbol{x}^{[i]}$ as $\bar{f}(\boldsymbol{x}^{[i]}) = 1, 2, 3, 4$ for $i = 1, \ldots, 4$ and yields $\mathbb{E}[\ell_{\mathrm{zo}}(\bar{f}(\boldsymbol{X}), Y)] = 0.5$. However, a classifier based on a threshold labeling with OT parameters $\boldsymbol{t}$ can achieve $\mathbb{E}[\ell_{\mathrm{zo}}(h_{\mathrm{thr}}(\bar{g}(\boldsymbol{X}); \boldsymbol{t}), Y)] = 0.55$, which is higher than $0.5$, at best.*

*Note that this example, showing that a classifier of 1DT-based methods can be sub-optimal, does not deny the practical utility of 1DT-based methods. Using a simple 1DT model that will be easy to learn may reduce the resulting generalization gap in a finite-size sample situation.* □

Also, the other remark is that the additional learning has a risk of enlarging the generalization gap. One can adjust the labeling function $h$ so that $h(\bar{g}(\boldsymbol{x}_i)) = y_i$ for every training example $i = 1, \ldots, n$ if allowing arbitrary formats and $y_{i_1} = y_{i_2}$ for any $i_1, i_2$ s.t. $\boldsymbol{x}_{i_1} = \boldsymbol{x}_{i_2}$, but the resulting classifier $h \circ \bar{g}$ would have quite low generalization performance. On the other hand, we here consider the additional learning of the labeling function among the class of threshold labelings. A threshold labeling has up to $(K-1)$ decision boundaries, that is, it is strictly restricted, and we expect that the degree of the generalization gap will not differ much with any threshold labelings.

## 4.2 Trouble with Brute-Force Search

In preparation for calculating the OT parameters, sort $\{(\bar{g}(\boldsymbol{x}_i), y_i)\}_{i=1}^n$ in the ascending order of $\{\bar{g}(\boldsymbol{x}_i)\}_{i=1}^n$, and represent the result as $\{(\bar{g}'_i, y'_i)\}_{i=1}^n$ with the modified index $i$ so that $\bar{g}'_1 \leq \cdots \leq \bar{g}'_n$. The sorting operation typically costs quasi-linear order $O(n \log n)$ computation loads in average (e.g., merge, heap, and quick sorts).

Assuming that the threshold parameter $t_k$ belongs to an interval $[\bar{g}'_{i-1}, \bar{g}'_i)$, the objective function $R(\boldsymbol{t}; \ell, \bar{g}, \mathcal{D}_n)$ with fixed $t_1, \ldots, t_{k-1}, t_{k+1}, \ldots, t_{K-1}$ remains the same value wherever $t_k$ is. Therefore, the OT parameters can be obtained by

$$\min_{\boldsymbol{t}} R(\boldsymbol{t}; \ell, \bar{g}, \mathcal{D}_n), \quad \text{subject to } t_1, \ldots, t_{K-1} \in \{c_i(\bar{g}, \mathcal{D}_n)\}_{i=1}^{n+1}, \tag{13}$$

with considering only the endpoints and midpoints

$$c_i(\bar{g}, \mathcal{D}_n) := \begin{cases} -\infty & \text{for } i = 1, \\ (\bar{g}'_{i-1} + \bar{g}'_i)/2 & \text{for } i = 2, \ldots, n, \\ +\infty & \text{for } i = n+1 \end{cases} \tag{14}$$

(in total $(n+1)$) as the candidates. Also, since $R(\boldsymbol{t}; \ell, \bar{g}, \mathcal{D}_n)$ is invariant with respect to the permutation of the threshold parameters $\boldsymbol{t}$, the search area of (13) can be further restricted to the ordered ones. Namely, we can incorporate the condition $t_1 \leq \cdots \leq t_{K-1}$ into (13). However, even under such restrictions, the problem (13) has $\binom{n+K-1}{n}$ candidates for the minimizer (recall the combination with repetition), where $\binom{n+K-1}{n} = O(n^{K-1})$ when $n \gg K$, and a solution based on the brute-force search takes a seriously high computation cost when the sample size $n$ is large.

## 4.3 Proposal of Independently Optimized Threshold (IOT) Labeling

Therefore, we propose to learn the threshold parameters independently through relaxed problems of the problem (13) (Algorithm 1), for higher computational efficiency; we call a threshold labeling consisting of threshold parameters obtained by Algorithm 1 the IOT labeling. The algorithm is developed according to the conditional relation

$$R(\boldsymbol{t}; \ell, \bar{g}, \mathcal{D}_n) = \sum_{k=1}^{K-1} R_k(t_k; \ell, \bar{g}, \mathcal{D}_n) - \underbrace{\sum_{k=2}^{K-1} R_k(+\infty; \ell, \bar{g}, \mathcal{D}_n)}_{\text{independent on } \boldsymbol{t}} \quad \text{if } t_1 \leq \cdots \leq t_{K-1} \tag{15}$$

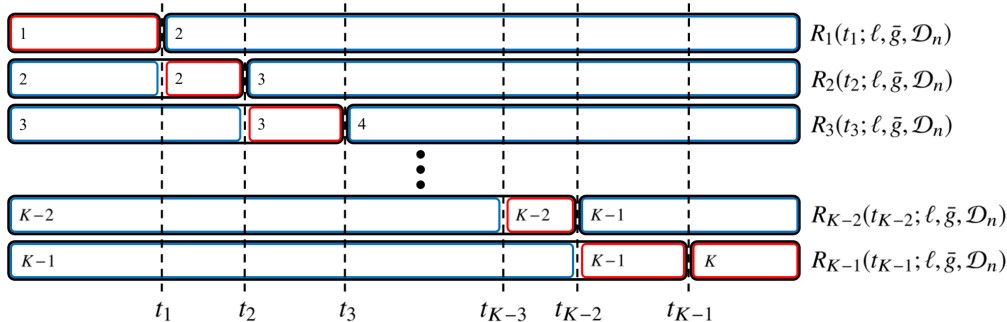

Figure 1: This figure is to help understanding of the equation (15). The 1DT $\bar{g}_i'$ located in the box marked with $k$ is labeled as $k$, and the corresponding task loss is $\ell(k, y_i')$.

---

**Algorithm 1:** To determine the threshold parameters for the IOT labeling[6]

---

**Input:** Task loss $\ell$, learned 1DT $\bar{g}$, and training data $\mathcal{D}_n = \{(\boldsymbol{x}_i, y_i)\}_{i=1}^n$.

/* Preparation. */

**1** Sort $\{(\bar{g}(\boldsymbol{x}_i), y_i)\}_{i=1}^n$ in the ascending order of $\{\bar{g}(\boldsymbol{x}_i)\}_{i=1}^n$, and represent the result as $\{(\bar{g}_i', y_i')\}_{i=1}^n$ with the modified index $i$ so that $\bar{g}_1' \leq \cdots \leq \bar{g}_n'$.

/* Narrow down candidates for the threshold parameters. */

**2 for** $k = 1, \ldots, K-1$ **do**

**3**      Initialization: $R_{k,1} = 0$.

**4**      **for** $i = 1, \ldots, n$ **do**

**5**          Sequential update: $R_{k,i+1} = R_{k,i} + \ell(k, y_i') - \ell(k+1, y_i')$.

**6**      For $\{i_{k,1}, \ldots, i_{k,n_k}\} = \arg\min_i \{R_{k,i}\}_{i=1}^{n+1}$, let $\{c_{i_{k,j}}(\bar{g}, \mathcal{D}_n)\}_{j=1}^{n_k}$ be candidates for $t_k$.

/* Determine the threshold parameters. */

**7** Determine $\bar{t}_1$ so that $\bar{t}_1 \leq \bar{t}_2$ will be more possible to hold: $\bar{t}_1 = \min\{c_{i_{1,j}}(\bar{g}, \mathcal{D}_n)\}_{j=1}^{i_1}$.

**8 for** $k = 2, \ldots, K-1$ **do**

**9**      Determine $\bar{t}_k$ so that $\bar{t}_{k-1} \leq \bar{t}_k$ if possible and $\bar{t}_k \leq \bar{t}_{k+1}$ will be more possible to hold:

         **if** $c_{i_{k,n_k}}(\bar{g}, \mathcal{D}_n) \geq \bar{t}_{k-1}$ **then**

            $\bar{t}_k = \min\{c_{i_{k,j}}(\bar{g}, \mathcal{D}_n) \mid c_{i_{k,j}}(\bar{g}, \mathcal{D}_n) \geq \bar{t}_{k-1}\}_{j=1}^{n_k}$.

         **else**

            $\bar{t}_k = c_{i_{k,n_k}}(\bar{g}, \mathcal{D}_n)$.

**Output:** Threshold parameters $\bar{\boldsymbol{t}} = (\bar{t}_1, \ldots, \bar{t}_{K-1})$.

---

with the functions $R_k$, $k = 1, \ldots, K-1$ defined by

$$R_k(t; \ell, \bar{g}, \mathcal{D}_n) := \frac{1}{n} \sum_{i=1}^n \ell(h_{\text{thr}}(\bar{g}(\boldsymbol{x}_i); (\cdots, -\infty, \underbrace{t}_{k\text{-th}}, +\infty, \cdots)), y_i), \tag{16}$$

which is an empirical task risk when it labels $\bar{g}'(\boldsymbol{x}_i) < t$ as $k$ and $\bar{g}'(\boldsymbol{x}_i) \geq t$ as $(k+1)$; see Figure 1, where the summation of the red parts implies the left-hand side term of (15), and the summation of the blue parts implies the latter term of the right-hand side of (15). This relation implies the equivalence between minimizing the empirical task risk $R(t; \ell, \bar{g}, \mathcal{D}_n)$ and minimizing $R_k(t_k; \ell, \bar{g}, \mathcal{D}_n)$, $k = 1, \ldots, K-1$ independently for each $k$ when the threshold parameters satisfy the order condition ($t_1 \leq \cdots \leq t_{K-1}$). Therefore, Algorithm 1 independently solves the latter relaxed subproblems associated with each threshold parameter.

The for-loop in Line 2 of Algorithm 1 can be parallel-processed. The computation cost required in addition to the sorting operation of Algorithm 1 is $O(n)$ less than $O(n \log n)$ in the training sample size $n$. Furthermore, we could prove that the IOT labeling becomes the OT labeling, as long as the obtained threshold parameters $\bar{\boldsymbol{t}}$ eventually follow the appropriate order $\bar{t}_1 \leq \cdots \leq \bar{t}_{K-1}$.

**Theorem 3.** *For any task loss $\ell$, 1DT $\bar{g}$, and data $\mathcal{D}_n$, the threshold parameters $\boldsymbol{t} = \bar{\boldsymbol{t}}$ obtained by Algorithm 1 minimize $R(\boldsymbol{t}; \ell, \bar{g}, \mathcal{D}_n)$ if they satisfy the order condition $\bar{t}_1 \leq \cdots \leq \bar{t}_{K-1}$.*

It would be difficult to judge before the learning whether the resulting threshold parameters follow the appropriate order, since the threshold parameters depend on the learned 1DT $\bar{g}$ whose theoretical properties are little known as reviewed in Section 3. The condition of the above theorem can be easily checked thanks to the important property that the computational load is not so large. Also, we expect that the condition will be met for a typical task loss $\ell$, surrogate loss $\phi$, and ordinal data $\mathcal{D}_n$, and succeeded learning of the 1DT; see the experimental verification in Section 5.

Note that when the order condition $\bar{t}_1 \leq \cdots \leq \bar{t}_{K-1}$ does not hold, Algorithm 1 has no performance guarantee. We set $\bar{t}_k = c_{i_{k,n_k}}(\bar{g}, \mathcal{D}_n)$ if $c_{i_{k,n_k}}(\bar{g}, \mathcal{D}_n) < \bar{t}_{k-1}$ in Line 9, so as to mitigate the influence of the violation of the order condition $\bar{t}_{k-1} \leq \bar{t}_k$. One can select another option such as $\bar{t}_k = \bar{t}_{k-1}$ (for that setting, the statement of Theorem 3 needs to be modified).

### 4.4 Relationship to Previous Works

Most studies on 1DT-based methods discuss the learning procedure, especially various options for the surrogate loss function. On the other hand, there are several previous works (Herbrich et al., 1999; Lin & Li, 2006) that have proposed labeling procedures different from the popular labeling procedures reviewed in Section 3 (albeit their proposed labeling procedures may be minor in that ideas for those have not been widely adopted in subsequent works). We here review the discussion of these previous works, and describe the relationship between their methods (especially the one by (Lin & Li, 2006)) and the IOT labeling.

Herbrich et al. (1999) considered pairwise ranking based on the 1DT of the explanatory variables and a hinge-type loss, under the task with the zero-one task loss. Their method (Herbrich et al., 1999, (12)) determines the threshold parameters on the basis of a unique criterion that emphasizes the shape of the used hinge-type loss and is different from the empirical task risk and empirical surrogate risk, and their threshold labeling has no optimality guarantee in minimization of the empirical task risk.

Lin & Li (2006) considered three methods for determining the threshold parameters; see (4), (7), and (8) of the reference. The third method is developed for the AT surrogate loss function with $\varphi(u) = e^{-u}$ and is the same as the MT and ST labelings in this paper. Also, one can understand that (7) tries to minimize (an upper bound of) the empirical surrogate risk based on the IT surrogate loss function with $\varphi(u) = e^{-u}$ and ordering constraint of the bias parameters, and the second labeling procedure also can be seen to follow the idea of the MT and ST labelings. Finally, the objective function (4) for the first method is defined as the empirical task risk with the absolute deviation task loss. However, they did not mention the relevance between the objective function and the task under consideration. Although Lin & Li (2006) presented a description on the threshold parameters determination based on the zero-one task loss at the part following (4), they tried only the threshold parameters determined by minimizing the empirical task risk with the absolute deviation task loss when they considered the task with the zero-one task loss in their experiments. The main claim of our consideration in Section 4 is that the threshold parameters should be determined via minimizing the (empirical) task risk for the task under consideration. Therefore, our consideration in Section 4 can be interpreted as a variant of the method (Lin & Li, 2006, (4)) with respect to the relevance between the objective function for determining the threshold parameters and the task under consideration. Also, Lin & Li (2006) described that (4) can be solved by dynamic programming, although they did not describe a more explicit procedure. Algorithm 1 may be similar to the one that they envisioned, but this paper presents the explicit procedure for a generic task loss as Algorithm 1 and reveals an optimality condition of that procedure in Theorem 3.

---

[6]We need to calculate $R_k(t; \ell, \bar{g}, \mathcal{D}_n)$, $k = 1, \ldots, K-1$ only for $t \in \{c_i(\bar{g}, \mathcal{D}_n)\}_{i=1}^{n+1}$, and so we simplified their notation to $R_{k,i}$. The resulting threshold parameters do not vary under the linear transformation of $\{R_{k,i}\}$. We thus initialized $R_{k,1}$ with 0 and omitted the averaging operation in Algorithm 1. These objects have the relationship,

$$R_{k,i} = n \{R_k(c_i(\bar{g}, \mathcal{D}_n); \ell, \bar{g}, \mathcal{D}_n) - R_k(-\infty; \ell, \bar{g}, \mathcal{D}_n)\} \text{ for } k = 1, \ldots, K-1, \ i = 1, \ldots, n+1. \tag{17}$$

## 5 Numerical Experiments

### 5.1 Purposes

We took numerical experiments to answer the questions,

- whether Algorithm 1 for the IOT labeling serves appropriately ordered threshold parameters, which is related to whether the IOT labeling improves the training task risk for 1DT-based methods,

- whether a modified threshold method with the IOT labeling yields better practical classification performance (i.e., smaller test task risk) than existing 1DT-based methods using other labelings,

- whether Algorithm 1 for the IOT labeling is computationally feasible.

We will present positive answers for the respective questions in Sections 5.3.1, 5.3.2, and 5.3.3.

### 5.2 Settings

#### 5.2.1 Datasets and Preprocessing

In the experiments, we used the five ***various-domain datasets***, LEV (lectures evaluation), ERA (employee rejection/acceptance), SWD (social workers decision), WQR (winequality-red), and CAR (car evaluation) datasets, and the three ***face-age datasets***, MORPH-2 (MORPH Album2), CACD, and AFAD datasets (Ricanek & Tesafaye, 2006; Chen et al., 2014; Niu et al., 2016). The main reason why we used the various-domain and face-age datasets is respectively to experiment with many datasets in various domains and to confirm whether the proposed method achieves the performance competitive to the state-of-the-art method in a modern application. For most of the experimental settings, such as preprocessing of datasets, task, and evaluation, we referred to those of the previous study (Cao et al., 2020).[7]

For the various-domain datasets, we selected those with the total sample size $n_{\text{tot}} \geq 1000$ from datasets that Gutierrez et al. (2015) used as OR datasets, and used them following the usage in (Gutierrez et al., 2015). One can obtain the various-domain datasets from a researchers' site (`http://www.uco.es/grupos/ayrna/orreview`) of (Gutierrez et al., 2015), or our GitHub repository (`https://github.com/Anonymous`). We purchased the MORPH-2 dataset at (`https://ebill.uncw.edu/C20231_ustores/web/`) and preprocessed it so that the face spanned the whole image with the nose tip, which was located by facial landmark detection (Sagonas et al., 2016), at the center by using `EyepadAlign` function by Raschka (2018). While this dataset contains 55,134 facial images with ages from 16 to 77, we used 55,013 images with ages from 16 to 70. The CACD dataset can be downloaded from (`https://bcsiriuschen.github.io/CARC/`). We preprocessed this dataset similarly to the MORPH-2 dataset. Since the CACD dataset collects images from the Internet using computer vision techniques, it includes some facial images inappropriate for our consideration. Excluding images, in which no face or more than two faces were detected in the preprocessing, from the original 163,446 images, we used 159,402 facial images in the age range of 14–62 years. For the AFAD dataset obtainable at (`https://github.com/afad-dataset/tarball`), because faces in its images were already centered, we took no further preprocessing, and used its 164,418 images with ages 15–40. For these face-age datasets, we treated the age rank as the target variable.

For the various-domain datasets, we randomly divided each dataset into $72\,\%$ training, $8\,\%$ validation, and $20\,\%$ test sets. For the face-age datasets, we resized all images to $128 \times 128 \times 3$ pixels (3 stems from RGB channels) and randomly divided each dataset into $72\,\%$ training, $8\,\%$ validation, and $20\,\%$ test sets, and the training phase used images randomly cropped with the size of $120 \times 120 \times 3$ pixels as input to improve the stability of the model against the difference of facial positions, and validation and test phases used images center-cropped to the same size, following (Cao et al., 2020)'s procedures.

---

[7]For the face-age datasets, we used a part of codes published in (`https://github.com/Raschka-research-group/coral-cnn`) by Cao et al. (2020), but results of our reproduction of their method differ from theirs mainly because we changed a learning rate from $5 \times 10^{-5}$ to $10^{-3}$. See (`https://github.com/Anonymous`) for used program codes.

Table 2: It shows basic dataset properties, the total sample size $n_{\text{tot}}$, the dimension $d$ of the explanatory variables, and the number $K$ of classes of the target variable, of all the used datasets.

|  | LEV | ERA | SWD | WQR | CAR | MORPH-2 | CACD | AFAD |
|---|---|---|---|---|---|---|---|---|
| $n_{\text{tot}}$ | 1000 | 1000 | 1000 | 1599 | 1728 | 55013 | 159402 | 164418 |
| $d$ | 4 | 4 | 10 | 11 | 21 | $128^2 \times 3$ | $128^2 \times 3$ | $128^2 \times 3$ |
| $K$ | 5 | 9 | 4 | 6 | 4 | 55 | 49 | 26 |

We summarized basic dataset properties, the total sample size $n_{\text{tot}}$, the dimension $d$ of the explanatory variables, and the number $K$ of classes of the target variable, of all the used datasets in Table 2.

### 5.2.2 Tasks and Methods

We considered two popular tasks in the experiments: minimization of the task risk for the absolute deviation loss $\ell_{\text{ad}}(j, k) = |j - k|$ (we call **Task-A**) and that for the squared loss $\ell_{\text{sq}}(j, k) = (j - k)^2$ (we call **Task-S**).

For the various-domain datasets, we applied a 1DT model based on a 4-layer fully-connected neural network, in which every hidden layer has 20 nodes activated with the sigmoid function in addition to bias nodes. Also, for the face-age datasets, we applied a 1DT model based on ResNet-34 (He et al., 2016), a modern CNN architecture, following (Cao et al., 2020)'s implementation. It modifies a fully-connected (the number of classes)-output final layer of the conventional ResNet-34 to a fully-connected 1-output layer.

Our tried loss functions are the SVOR loss (3), NLL (7)[8] and ANLCL (8) losses for the statistical model (6) with the sigmoid function $\sigma = \sigma_{\text{olr}}$, and AD loss $\phi = \phi_{\text{ad}}$, each of which is a representative instance of bias-parametric non-statistical, bias-parametric statistical, and bias-nonparametric losses used in the 1DT-based methods. Also, in another taxonomy, SVOR is an IT loss, and ANLCL is an AT loss; see Table 1.

For the learning procedure of the 1DT model and the bias parameters with bias-parametric losses, we examined two ways, with and without the ordering constraint on the bias parameters.

Without the ordering constraint on the bias parameters, we tried the MT, ST, and IOT labelings along with the SVOR loss, the MT (equal to ST), LB, and IOT labelings along with the NLL loss, the MT, ST, LB, and IOT labelings along with the ANLCL loss, the NNT and IOT labelings along with the AD loss. When the NLL loss is used, learned bias parameters will be ordered (otherwise, the objective function takes Not a Number), and the MT and ST labelings bring the same result. With the ordering constraint, the MT and ST labelings become the same.

Note that Cao et al. (2020) declare that their method (ANLCL, Off, ST), which is a combination of the ANLCL loss and the ST labeling without the ordering constraint on the bias parameters, is the state-of-the-art method for the face-age datasets in 2020.

### 5.2.3 Training and Evaluation

During the validation and test phases, models are evaluated based on the mean absolute error (**MAE**) and the root of the mean squared error (**RMSE**), which are defined for a classifier $f$ and $m$ used data points as $\frac{1}{m} \sum_{i=1}^{m} \ell_{\text{ad}}(f(\boldsymbol{x}_i), y_i)$ and $\{\frac{1}{m} \sum_{i=1}^{m} \ell_{\text{sq}}(f(\boldsymbol{x}_i), y_i)\}^{1/2}$, for the Task-A and Task-S. Here, the root operation of the RMSE is only for adjusting the scale of the error and does not affect our discussion related to the optimality of IOT and so on.

We ran 20 trials with randomly-set different divisions of training, validation, and test sets and initial parameters of the network. In each trial, we trained the network using Adam of the learning rate $10^{-3}$ with `DataLoader` of `batch_size` 256 for the face-age datasets or integer $\leq 0.072 \cdot n_{\text{tot}}$ for the various-domain datasets and `num_workers` 16 (in `Pytorch`) as an optimization procedure for 200 epochs. For the learning rate, although Cao et al. (2020) used $5 \times 10^{-5}$, we selected the one with the best validation result for most combinations of the datasets, methods, and tasks, from $\{10^{-5}, 10^{-4}, \cdots, 10^{-1}\}$ in our preliminary experi-

---

[8]For numerical stability (to avoid $\log(0)$), we used an approximation of NLL loss in which the logarithmic function $\log(\cdot)$ of (7) is replaced to $\log(\cdot + 10^{-8})$ in the experiments.

ments. Additionally, for a method using the IOT labeling, we calculated the threshold parameters according to Algorithm 1 at the end of every training epoch.

The above errors were evaluated on the validation set at the end of every training epoch, and then we adopted a model at the timing with the smallest error among the obtained validation error sequences as the test model.

We judge the significance on the classification performance of the labeling function by the one-sided Wilcoxon rank sum test with $p$-value 0.05 based on errors for 20 trials of methods using different labeling functions, in each combination of the dataset, error, and surrogate loss function.

### 5.3 Results

#### 5.3.1 Order of Threshold Parameters obtained by Algorithm 1

In all the combinations of datasets, tasks, surrogate losses, and 20 trials, Algorithm 1 for the IOT labeling served appropriately ordered threshold parameters for a learned 1DT of the test model. Of course, it has no guarantee that the algorithm will serve ordered threshold parameters for an insufficiently-trained 1DT model, for example, a 1DT model with initial parameters. It, however, served ordered threshold parameters for a 1DT model after every epoch in most cases.

Figure 2 shows the learning curves of MAE and RMSE versus training epoch in a certain trial, as demonstration. In this trial, threshold parameters obtained by Algorithm 1 were ordered at all the epochs. Therefore, as shown by Theorem 3, one can see that using the IOT labeling improves the training task risk at each epoch (every blue curve is located below related red and green curves).

#### 5.3.2 Generalization Performance

Table 3 shows the mean and standard deviation of the errors, for the test model, evaluated on the test set. Theorem 3 and the fact that the threshold parameters obtained by Algorithm 1 were appropriately ordered do not directly imply the superiority relation of the test task risks of different methods based on different 1DT models. However, in many tried cases, the IOT labeling improved not only the MT, ST, and NNT labelings but also the LB labeling (in statistical methods) regarding the test task risk. Especially for the face-age datasets, modified threshold methods using the IOT labeling provided better performance than (Cao et al., 2020)'s method (ANLCL, Off, ST) that was the state-of-the-art in 2020. These results suggest the success of the IOT labeling in the subject (aiming for a better labeling procedure) of our research.

#### 5.3.3 Calculation Time

We are also interested in the computational efficiency of the IOT labeling. The IOT labeling is more disadvantageous in terms of the computational efficiency than other existing labelings as the training sample size increases. Thus, for the AFAD dataset with the largest sample size, we evaluated computation times required for the four methods with the MT, ST, LB, and IOT labelings, under the fixed setting with the ANLCL loss and without ordering constraint of bias parameters, and Task-A, in a certain trial (using different losses and randomness due to random seeding would little affect the computation time). Note that we experimented with program codes based on Python 3.8.12 and Pytorch 1.10.1 (see also footnote 7) under the computation environment with a CPU Intel Xeon Silver 4108 and a GPU GeForce RTX A6000.

The training procedure is common for all the labelings, and to update the network parameters using the optimization algorithm and all training data points, where we used Adam with `DataLoader` of `batch_size` 256 and `num_workers` 16 (in Pytorch) for optimization. The validation procedure is to evaluate the validation task risk $\frac{1}{m}\sum_{i=1}^{m}\ell_{\mathrm{ad}}(h(g(\boldsymbol{x}_i)), y_i)$ for a given 1DT model $g$ and a given labeling $h$. Since the MT, ST, and LB labelings are pre-designed they do not need additional computations to learn the labeling $h$, but the IOT labeling does (as Algorithm 1). Thus, for the IOT labeling, we further evaluated the computation time taken for Algorithm 1, where we used quick sort, which was the default as `sort` in Pytorch, for Line 1 of Algorithm 1.

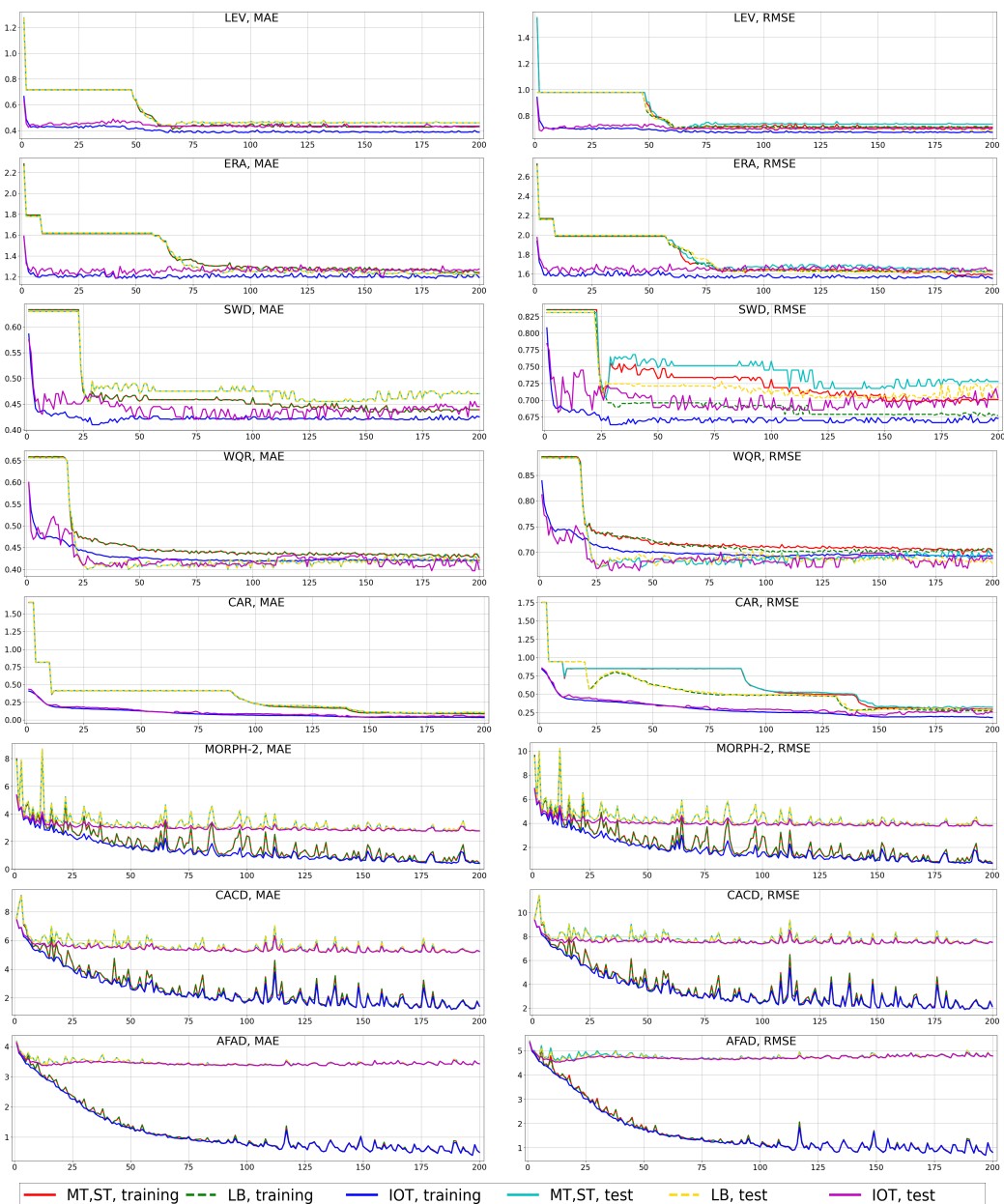

Figure 2: It shows the learning curves of MAE and RMSE evaluated on training and test sets in a certain trial versus training epoch, for methods with the ANLCL loss and the MT, LB, and IOT labelings and without the ordering constraint on the bias parameters, and for all the used datasets. In this trial, since the bias parameters were ordered consequently, the MT labeling and LB labeling for the Task-A were equal to the ST labeling (this result demonstrates Theorem 2 (iii) about the relation of the LB and ST labelings).

Table 4 shows the mean (over 200 epochs) of these computation times. The modified threshold method using the IOT labeling took just at most $\frac{51.666+7.387+4.309}{51.666+4.249} \approx 1.133$ times as long as a conventional threshold method even if we calculated a validation error every epoch, despite that the AFAD dataset has as many as $164,418 \times 0.72 \approx 118,000$ training data points. The ratio depends on the computational environments and was about 1.081 when using a GPU GeForce RTX 2080 Ti. These results would demonstrate the computational feasibility of the IOT labeling.

Table 3: It shows the mean (M) and standard deviation (S) of the test errors in the form '$M_S$'. For a method using a bias-parametric loss, we respectively mark (On) and (Off) under the loss if it adopts the ordering constraint of the bias parameters and does not. The smaller the mean of an error, the better that method is for that dataset and that task. In each block specified with the dataset, error, and loss, we highlighted in bold font the best results that were tie each other and superior to all other results with respect to the one-sided Wilcoxon rank sum test with a significance level of 0.05, if they exist. Also, we colored the best result in red for each combination of dataset and error.

| | Loss | Labeling | LEV | ERA | SWD | WQR | CAR | MORPH-2 | CACD | AFAD |
|---|---|---|---|---|---|---|---|---|---|---|
| **MAE for Task-A** | SVOR (On) | MT,ST | $.482_{023}$ | $1.262_{052}$ | $.465_{034}$ | $.462_{018}$ | $.130_{009}$ | $3.147_{067}$ | $6.776_{146}$ | $3.927_{091}$ |
| | | IOT | $\mathbf{.420_{039}}$ | $1.229_{057}$ | $\mathbf{.441_{029}}$ | $\mathbf{.447_{028}}$ | $\mathbf{.062_{011}}$ | $\mathbf{3.016_{029}}$ | $\mathbf{6.386_{078}}$ | $\mathbf{3.643_{016}}$ |
| | SVOR (Off) | MT | $.532_{028}$ | $1.280_{048}$ | $.453_{038}$ | $.465_{019}$ | $.130_{012}$ | $2.944_{029}$ | $5.280_{029}$ | $3.526_{031}$ |
| | | ST | $.532_{028}$ | $1.280_{048}$ | $.453_{038}$ | $.465_{019}$ | $.090_{010}$ | $2.944_{029}$ | $5.280_{029}$ | $3.514_{033}$ |
| | | IOT | $\mathbf{.418_{040}}$ | $\mathbf{1.226_{061}}$ | $.445_{027}$ | $\mathbf{.450_{028}}$ | $\mathbf{.060_{010}}$ | $\mathbf{2.910_{030}}$ | $\mathbf{5.242_{029}}$ | $\mathbf{3.401_{021}}$ |
| | NLL (On) | MT,ST | $.426_{037}$ | $1.269_{062}$ | $.448_{035}$ | $.448_{026}$ | $.078_{018}$ | $2.783_{019}$ | $5.146_{026}$ | $3.336_{015}$ |
| | | LB | $.426_{037}$ | $1.269_{062}$ | $.448_{035}$ | $.448_{026}$ | $.078_{018}$ | $2.783_{019}$ | $5.146_{026}$ | $3.336_{015}$ |
| | | IOT | $.421_{042}$ | $\mathbf{1.229_{054}}$ | $.438_{028}$ | $.450_{024}$ | $\mathbf{.054_{010}}$ | $2.777_{022}$ | $5.138_{026}$ | $3.335_{015}$ |
| | NLL (Off) | MT,ST | $.427_{042}$ | $1.268_{056}$ | $.447_{033}$ | $.445_{021}$ | $.079_{013}$ | $2.797_{015}$ | $5.158_{025}$ | $3.340_{019}$ |
| | | LB | $.427_{042}$ | $1.268_{056}$ | $.447_{033}$ | $.445_{021}$ | $.079_{013}$ | $2.797_{015}$ | $5.158_{025}$ | $3.340_{019}$ |
| | | IOT | $.420_{038}$ | $\mathbf{1.222_{061}}$ | $.442_{025}$ | $.447_{024}$ | $\mathbf{.054_{011}}$ | $2.788_{023}$ | $5.157_{025}$ | $3.335_{016}$ |
| | ANLCL (On) | MT,ST | $.422_{039}$ | $1.271_{054}$ | $.443_{035}$ | $.444_{021}$ | $.088_{011}$ | $2.756_{020}$ | $5.165_{030}$ | $3.368_{017}$ |
| | | LB | $.422_{039}$ | $1.271_{054}$ | $.443_{035}$ | $.444_{021}$ | $.088_{011}$ | $2.756_{020}$ | $5.165_{030}$ | $3.368_{017}$ |
| | | IOT | $.424_{045}$ | $\mathbf{1.219_{055}}$ | $.440_{035}$ | $.451_{029}$ | $\mathbf{.051_{009}}$ | $2.752_{017}$ | $5.162_{028}$ | $3.368_{015}$ |
| | ANLCL (Off) | MT | $.446_{048}$ | $1.273_{055}$ | $.462_{041}$ | $.444_{020}$ | $.100_{012}$ | $2.773_{025}$ | $5.159_{019}$ | $3.376_{015}$ |
| | | ST | $.446_{048}$ | $1.273_{055}$ | $.462_{041}$ | $.444_{020}$ | $.100_{012}$ | $2.773_{025}$ | $5.159_{019}$ | $3.376_{015}$ |
| | | LB | $.446_{048}$ | $1.273_{055}$ | $.462_{041}$ | $.444_{020}$ | $.100_{012}$ | $2.773_{025}$ | $5.159_{019}$ | $3.376_{015}$ |
| | | IOT | $\mathbf{.417_{036}}$ | $\mathbf{1.218_{058}}$ | $.442_{033}$ | $.449_{026}$ | $\mathbf{.049_{009}}$ | $2.769_{018}$ | $5.157_{019}$ | $3.369_{014}$ |
| | AD | NNT | $.455_{037}$ | $1.297_{049}$ | $.478_{024}$ | $.467_{023}$ | $.091_{011}$ | $2.811_{021}$ | $5.141_{035}$ | $3.337_{012}$ |
| | | IOT | $\mathbf{.422_{040}}$ | $\mathbf{1.230_{061}}$ | $\mathbf{.447_{021}}$ | $\mathbf{.446_{025}}$ | $\mathbf{.063_{011}}$ | $2.806_{025}$ | $5.147_{036}$ | $3.345_{017}$ |
| **RMSE for Task-S** | SVOR (On) | MT,ST | $.755_{023}$ | $1.640_{064}$ | $.706_{029}$ | $.738_{018}$ | $.450_{021}$ | $4.290_{096}$ | $8.701_{151}$ | $5.174_{121}$ |
| | | IOT | $\mathbf{.698_{044}}$ | $\mathbf{1.598_{056}}$ | $.692_{026}$ | $.729_{027}$ | $\mathbf{.268_{033}}$ | $\mathbf{4.059_{036}}$ | $\mathbf{8.294_{066}}$ | $\mathbf{4.825_{017}}$ |
| | SVOR (Off) | MT | $.792_{024}$ | $1.671_{071}$ | $.714_{040}$ | $.741_{017}$ | $.450_{027}$ | $4.015_{043}$ | $7.379_{048}$ | $4.779_{056}$ |
| | | ST | $.792_{024}$ | $1.671_{071}$ | $.714_{040}$ | $.741_{017}$ | $.306_{021}$ | $4.017_{045}$ | $7.379_{048}$ | $4.727_{040}$ |
| | | IOT | $\mathbf{.695_{047}}$ | $\mathbf{1.595_{056}}$ | $.694_{020}$ | $.728_{024}$ | $\mathbf{.260_{037}}$ | $\mathbf{3.984_{037}}$ | $\mathbf{7.329_{043}}$ | $\mathbf{4.566_{022}}$ |
| | NLL (On) | MT,ST | $.701_{045}$ | $1.634_{057}$ | $.694_{033}$ | $.722_{026}$ | $.302_{056}$ | $3.798_{049}$ | $7.340_{038}$ | $4.557_{022}$ |
| | | LB | $.711_{040}$ | $1.636_{051}$ | $.690_{025}$ | $.721_{028}$ | $.293_{042}$ | $3.795_{045}$ | $7.336_{041}$ | $\mathbf{4.509_{022}}$ |
| | | IOT | $.695_{046}$ | $1.608_{054}$ | $.696_{022}$ | $.728_{024}$ | $\mathbf{.242_{027}}$ | $3.787_{039}$ | $7.357_{043}$ | $4.509_{011}$ |
| | NLL (Off) | MT,ST | $.700_{048}$ | $1.638_{060}$ | $.705_{035}$ | $.723_{025}$ | $.285_{040}$ | $3.837_{026}$ | $7.333_{041}$ | $4.563_{026}$ |
| | | LB | $.710_{043}$ | $1.640_{056}$ | $.691_{027}$ | $.726_{027}$ | $.299_{024}$ | $3.839_{028}$ | $7.332_{043}$ | $4.527_{022}$ |
| | | IOT | $.697_{048}$ | $1.605_{057}$ | $.690_{023}$ | $.728_{023}$ | $\mathbf{.241_{029}}$ | $3.829_{029}$ | $\mathbf{7.292_{028}}$ | $4.515_{019}$ |
| | ANLCL (On) | MT,ST | $.698_{047}$ | $1.629_{055}$ | $.697_{032}$ | $.719_{022}$ | $.300_{022}$ | $3.773_{027}$ | $7.380_{032}$ | $4.585_{028}$ |
| | | LB | $.710_{042}$ | $1.633_{042}$ | $.695_{026}$ | $.722_{026}$ | $.285_{019}$ | $3.772_{027}$ | $7.372_{031}$ | $4.534_{025}$ |
| | | IOT | $.700_{047}$ | $\mathbf{1.595_{047}}$ | $.694_{029}$ | $.722_{023}$ | $\mathbf{.226_{021}}$ | $3.764_{024}$ | $7.402_{033}$ | $4.537_{016}$ |
| | ANLCL (Off) | MT | $.719_{050}$ | $1.628_{057}$ | $.722_{039}$ | $.722_{024}$ | $.320_{021}$ | $3.790_{030}$ | $7.370_{047}$ | $4.590_{033}$ |
| | | ST | $.719_{050}$ | $1.628_{057}$ | $.722_{039}$ | $.722_{024}$ | $.320_{021}$ | $3.790_{030}$ | $7.370_{047}$ | $4.590_{033}$ |
| | | LB | $.715_{041}$ | $1.635_{043}$ | $.689_{022}$ | $.723_{026}$ | $.289_{016}$ | $3.788_{027}$ | $7.359_{047}$ | $\mathbf{4.537_{021}}$ |
| | | IOT | $.697_{047}$ | $\mathbf{1.599_{051}}$ | $\mathbf{.691_{027}}$ | $.722_{026}$ | $\mathbf{.226_{025}}$ | $3.781_{026}$ | $7.376_{027}$ | $\mathbf{4.534_{017}}$ |
| | AD | NNT | $.720_{043}$ | $1.654_{043}$ | $.708_{024}$ | $.743_{021}$ | $.325_{026}$ | $3.844_{033}$ | $7.303_{036}$ | $4.571_{023}$ |
| | | IOT | $.707_{044}$ | $\mathbf{1.599_{055}}$ | $.698_{019}$ | $\mathbf{.725_{022}}$ | $\mathbf{.267_{024}}$ | $3.829_{038}$ | $7.287_{037}$ | $\mathbf{4.540_{018}}$ |

# 6 Conclusion and Future Prospect

We showed in Theorem 2 that not only the MT, ST, and NNT labelings but also the LB labeling is a threshold labeling in typical settings. In this study we considered the OT labeling to obtain higher classification

Table 4: It shows the mean calculation time for 1 epoch (in seconds), when using the ANLCL loss without the ordering constraint on the bias parameters, under the Task-A.

| Labeling Phase | all training | MT validation | ST validation | LB validation | IOT Algorithm 1 | IOT validation |
|---|---|---|---|---|---|---|
| Time | 51.666 | 4.679 | 4.249 | 4.686 | 7.387 | 4.309 |

performance than these threshold labelings used in the existing studies and proposed the IOT labeling as a more computationally efficient alternative labeling. Theorem 3 provides a condition for the IOT labeling to be the OT labeling. Experiments in this paper showed the satisfaction of the optimality condition, superior classification performance, and computational feasibility of the IOT labeling. On the ground of these consequences, we suggest a modified threshold method using the IOT labeling among the 1DT-based methods.

We are also interested in the selection of the learning procedure, especially the surrogate loss function, of the threshold method. One may be able to take systematic discussion on the goodness of the loss function by fixing components of the threshold method other than the loss function to the optimal ones. In such discussion, the OT and IOT labelings will serve as the optimal other components. This is a future prospect.

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

## A  Proof of Consistency of Statistical Methods

We here give proof of the theorem related to the interpretation of statistical loss functions.

*Proof of Theorem 1.* We can characterize the surrogate risk minimization for the NLL loss (7) as maximum likelihood estimation for the statistical model (6) for multi-class classification problem through the equation

$$
\begin{aligned}
\min_{g\in\mathcal{G}, \boldsymbol{b}\in\mathcal{B}} \mathbb{E}[\phi_{\mathrm{nll}}(g(\boldsymbol{X}), \boldsymbol{b}, Y)] &= \min_{g\in\mathcal{G}, \boldsymbol{b}\in\mathcal{B}} \mathbb{E}\left[\sum_{y=1}^{K} \Pr(y|\boldsymbol{X})\phi_{\mathrm{nll}}(g(\boldsymbol{X}), \boldsymbol{b}, y)\right] \\
&= \min_{g\in\mathcal{G}, \boldsymbol{b}\in\mathcal{B}} \mathbb{E}\left[-\sum_{y=1}^{K} \Pr(y|\boldsymbol{X})\log\hat{\Pr}(y|\boldsymbol{X}; g, \boldsymbol{b})\right].
\end{aligned}
\tag{18}
$$

According to the method of Lagrange multiplier, one solution of a point-wise (at each $\boldsymbol{X} = \boldsymbol{x}$) minimization problem

$$
\min_{\{\hat{\Pr}(k|\boldsymbol{x})\}_k} -\sum_{y=1}^{K} \Pr(y|\boldsymbol{x})\log\hat{\Pr}(y|\boldsymbol{x}), \quad \text{subject to } \sum_{y=1}^{K} \hat{\Pr}(y|\boldsymbol{x}) = 1
\tag{19}
$$

is $\hat{\Pr}(y|\boldsymbol{x}) = \Pr(y|\boldsymbol{x}) = \hat{\Pr}(y|\boldsymbol{x}; \tilde{g}, \tilde{\boldsymbol{b}})$, $y = 1, \ldots, K$, where the existence of such $\{\tilde{g}(\boldsymbol{x}), \tilde{\boldsymbol{b}}\}$ is assumed in the statement of the theorem. This solution applies for any $\boldsymbol{x} \in \mathbb{R}^d$, and one can see that a solution of (18) is $\{\tilde{g}, \tilde{\boldsymbol{b}}\}$, which completes the proof of the statement for the NLL loss (7).

Also, for the ANLCL loss (8), we can provide the following characterization:

$$
\begin{aligned}
&\mathbb{E}[\phi_{\mathrm{anlcl}}(g(\boldsymbol{X}), \boldsymbol{b}, Y)] \\
&= \mathbb{E}\left[-\sum_{y=1}^{K} \Pr(y|\boldsymbol{X})\left\{\sum_{k=1}^{y-1} \log\left\{1 - \hat{\Pr}(Y \leq k|\boldsymbol{X}; g, \boldsymbol{b})\right\} + \sum_{k=y}^{K-1} \log\left\{\hat{\Pr}(Y \leq k|\boldsymbol{X}; g, \boldsymbol{b})\right\}\right\}\right] \\
&= -\sum_{y=1}^{K-1} \mathbb{E}\left[\Pr(Y \leq y|\boldsymbol{X})\log\hat{\Pr}(Y \leq y|\boldsymbol{X}; g, \boldsymbol{b}) + \{1 - \Pr(Y \leq y|\boldsymbol{X})\}\log\left\{1 - \hat{\Pr}(Y \leq y|\boldsymbol{X}; g, \boldsymbol{b})\right\}\right],
\end{aligned}
\tag{20}
$$

where $\hat{\Pr}(Y \leq y|\boldsymbol{X}; g, \boldsymbol{b}) := \sum_{k=1}^{y} \hat{\Pr}(k|\boldsymbol{X}; g, \boldsymbol{b})$ and the expectation value $\mathbb{E}[\cdot]$ is taken for $\boldsymbol{X}$. On the ground of the binary version, '$y$ or less' vs. 'more than $y$' ($y = 1, \ldots, K-1$), of (19), one can prove the statement similarly. □

One may consider the (our-called) immediate negative log cumulative likelihoods (INLCL) loss function,

$$
\phi_{\mathrm{inlcl}}(g(\boldsymbol{x}), \boldsymbol{b}, y) := -\log(\sigma(g(\boldsymbol{x}) - b_{y-1})) - \log(\sigma(b_y - g(\boldsymbol{x}))),
\tag{21}
$$

which follows the framework of the IT loss in Table 1. However, it is difficult to characterize the surrogate risk minimization with the INLCL loss as a problem with a known solution unlike those for the NLL and ANLCL losses, and the optimality condition for the INLCL loss is unknown.

## B  Proof of Relationships between Labeling Functions

This section provides proofs of Theorem 2 and Corollary 1 regarding the relationships between the LB and threshold labelings. Propositions 1 and 2 would be trivial, so we omit proofs of them.

First, we prove Theorem 2.

*Proof of Theorem 2.* We introduce the functions

$$
L_j(u) := \sum_{k=1}^{K} \{\sigma(\bar{b}_k - u) - \sigma(\bar{b}_{k-1} - u)\}\ell(j, k) = \ell(j, K) + \sum_{k=1}^{K-1} \sigma(\bar{b}_k - u)\{\ell(j, k) - \ell(j, k+1)\},
\tag{22}
$$

with $j = 1, \ldots, K$, where the equation holds since $\bar{b}_0 = -\infty$, $\bar{b}_K = +\infty$, $\sigma(-\infty) = 0$, and $\sigma(+\infty) = 1$. The classifier based on the LB labeling, $f(\boldsymbol{x}) = \arg\min_{j \in [K]} \sum_{k=1}^{K} \hat{\Pr}(k|\boldsymbol{x}; \bar{g}, \bar{\boldsymbol{b}}) \ell(j, k)$, is equal to $\arg\min_{j \in [K]} L_j(\bar{g}(\boldsymbol{x}))$. According to Proposition 1, the LB labeling is a certain threshold labeling if and only if $\arg\min_{j \in [K]} \{L_j(u_1)\}_{j=1}^{K} \leq \arg\min_{j \in [K]} \{L_j(u_2)\}_{j=1}^{K}$ for any $u_1, u_2 \in \mathbb{R}$ such that $u_1 \leq u_2$. The latter condition holds if the situation

$$L_k(u) > L_l(u) \text{ for } u \in (s_1, s_2) \text{ and } L_k(u) < L_l(u) \text{ for } u \in (s_2, s_3) \text{ with } k < l, \ s_1 < s_2 < s_3 \tag{23}$$

does not occur. In the following we assume $k < l$ for the indices $k, l \in [K]$.

**Proof of (i).** Under the assumption described in the statement of the theorem, the difference

$$L_k(u) - L_l(u) = \underbrace{\{\ell(k, K) - \ell(l, K)\}}_{\substack{\text{non-negative} \\ \text{constant}}} + \sum_{j=1}^{K-1} \underbrace{\sigma(\bar{b}_j - u)}_{\substack{\text{non-negative} \\ \text{non-increasing}}} \underbrace{\{\ell(k, j) - \ell(k, j+1) - \ell(l, j) + \ell(l, j+1)\}}_{\substack{\text{non-positive} \\ \text{constant}}} \tag{24}$$

is non-decreasing with respect to $u$. Thus, $L_k(u) \leq L_l(u)$ for $u \leq p$ and $L_k(u) \geq L_l(u)$ for $u \geq p$ for a some point $p$, $L_k(u) \leq L_l(u)$ for any $u$, or $L_k(u) \geq L_l(u)$ for any $u$, which implies that the above-mentioned situation (23) does not occur. Note that, for the instances $\ell = \ell_{\text{ad}}, \ell_{\text{sq}}$, one has that

$$\ell_{k,l}(j) = \ell(k, j) - \ell(k, j+1) - \ell(l, j) + \ell(l, j+1) = \begin{cases} -2 \cdot \mathbf{1}_{k \leq j \leq l-1} & \text{for } \ell = \ell_{\text{ad}}, \\ 2(k - l), & \text{for } \ell = \ell_{\text{sq}}. \end{cases} \tag{25}$$

This completes the proof of the statement (i).

**Proof of (ii).** For $\ell = \ell_{\text{zo},c}$ with $c \in [0, \lfloor K/2 \rfloor)$ where $\ell_{\text{zo}} = \ell_{\text{zo},0}$, the function $L_j(u)$ reduces to

$$L_j(u) = 1 - \{\sigma(b_j - u) - \sigma(a_j - u)\}, \tag{26}$$

with $a_j := \bar{b}_{\max\{0, j-c\}}$ and $b_j := \bar{b}_{\min\{j+c, K\}}$, where $a_j < b_j$. Lemma 1 (described after the proof of Theorem 2) shows the shape of the function $L_j(u)$: Under the assumption of Theorem 2 (ii), $L_j(u)$ is minimized at $u = (a_j + b_j)/2 := c_j$, symmetric in $u$ around $u = c_j$, non-increasing in $u$ for $u < c_j$, and non-decreasing in $u$ when $u > c_j$, from Lemma 1 (i) and (ii). Also, assuming that $c_j$ is fixed, then $L_j(u)$ is non-decreasing in $b_j - a_j$, from Lemma 1 (iii).

When $b_k - a_k = b_l - a_l$, the translated two curves $L_k(u)$ and $L_l(u)$ have just one intersection point at $u = (c_k + c_l)/2$, and it holds that $L_k(u) \leq L_l(u)$ for $u \leq (c_k + c_l)/2$ and $L_k(u) \geq L_l(u)$ for $u \geq (c_k + c_l)/2$. Therefore, the situation (23) does not occur if $b_k - a_k = b_l - a_l$.

Then, assume $b_k - a_k < b_l - a_l$ (the following proof strategy for this setting can be applied to the other setting $b_k - a_k > b_l - a_l$). In this setting, $L_k(u) > L_l(u)$ for $u \geq c_l$ due to the shape of the functions $L_k$ and $L_l$. Also, within $[c_k, c_l]$, they can have one intersection point $p$ at most such that $L_k(u) \leq L_l(u)$ for $u \in [c_k, p]$ and $L_k(u) \geq L_l(u)$ for $u \in [p, c_l]$, since $L_k(u)$ and $L_l(u)$ are respectively non-decreasing and non-increasing in $u$. Therefore, the situation (23) can be satisfied only in such a situation that there exists a point $p$ satisfying

$$L_k(p) = L_l(p), \ L_k'(p) < L_l'(p), \text{ and } p \leq c_k. \tag{27}$$

The existence of such a point $p$ implies that

$$\frac{\sigma'(a_k - p) - \sigma'(b_k - p)}{\sigma(a_k - p) - \sigma(b_k - p)} < \frac{\sigma'(a_l - p) - \sigma'(b_l - p)}{\sigma(a_l - p) - \sigma(b_l - p)} \text{ with } a_k \leq a_l, \ b_k \leq b_l, \ a_k \leq b_k, \ a_l \leq b_l, \ p \leq c_k. \tag{28}$$

However, the assumption that $\frac{\sigma'(v_1) - \sigma'(v_2)}{\sigma(v_1) - \sigma(v_2)}$ is non-increasing in $v_1$ with fixed $v_2$ and in $v_2$ with fixed $v_1$ when $v_1 < v_2$ shows that

$$\frac{\sigma'(a_k - p) - \sigma'(b_k - p)}{\sigma(a_k - p) - \sigma(b_k - p)} \geq \frac{\sigma'(a_k - p) - \sigma'(b_l - p)}{\sigma(a_k - p) - \sigma(b_l - p)} \geq \frac{\sigma'(a_l - p) - \sigma'(b_l - p)}{\sigma(a_l - p) - \sigma(b_l - p)}, \tag{29}$$

which contradicts to the equation (28). Therefore, the situation (23) does not occur also when $b_k - a_k < b_l - a_l$.

Note that, especially when $\sigma = \sigma_{\text{olr}}$, one can show that

$$\frac{\sigma'(v_1) - \sigma'(v_2)}{\sigma(v_1) - \sigma(v_2)} = \frac{\sigma_{\text{olr}}(v_1)(1 - \sigma_{\text{olr}}(v_1)) - \sigma_{\text{olr}}(v_2)(1 - \sigma_{\text{olr}}(v_2))}{\sigma_{\text{olr}}(v_1) - \sigma_{\text{olr}}(v_2)} = 1 - \{\sigma_{\text{olr}}(v_1) + \sigma_{\text{olr}}(v_2)\}, \tag{30}$$

is decreasing in $v_1$ with fixed $v_2$ and in $v_2$ with fixed $v_1$. Moreover, when $\sigma = \sigma_{\text{gpor}}$, one has that

$$\frac{\sigma'(v_1) - \sigma'(v_2)}{\sigma(v_1) - \sigma(v_2)} \propto \frac{e^{-v_1^2/2} - e^{-v_2^2/2}}{\sigma_{\text{gpor}}(v_1) - \sigma_{\text{gpor}}(v_2)} := f_1(v_1, v_2), \tag{31}$$

that the derivative of $f_1(v_1, v_2)$ with respect to $v_1$,

$$\frac{\partial}{\partial v_1} f_1(v_1, v_2) = \frac{-v_1 e^{-v_1^2/2}\{\sigma_{\text{gpor}}(v_1) - \sigma_{\text{gpor}}(v_2)\} - \left(e^{-v_1^2/2} - e^{-v_2^2/2}\right)\frac{1}{\sqrt{2\pi}}e^{-v_1^2/2}}{\{\sigma_{\text{gpor}}(v_1) - \sigma_{\text{gpor}}(v_2)\}^2} \tag{32}$$

has the same sign as

$$f_2(v_1, v_2) := -v_1\{\sigma_{\text{gpor}}(v_1) - \sigma_{\text{gpor}}(v_2)\} - \left(\frac{1}{\sqrt{2\pi}}e^{-v_1^2/2} - \frac{1}{\sqrt{2\pi}}e^{-v_2^2/2}\right), \tag{33}$$

and that the derivative of $f_2(v_1, v_2)$ with respect to $v_2$ is

$$\frac{\partial}{\partial v_2} f_2(v_1, v_2) = (v_1 - v_2)\frac{1}{\sqrt{2\pi}}e^{-v_2^2/2}. \tag{34}$$

Since $\frac{\partial}{\partial v_2} f_2(v_1, v_2) < 0$ when $v_1 < v_2$ and $f_2(v_1, v_1) = 0$, it holds that $f_2(v_1, v_2)$, which has the same sign as $\frac{\partial}{\partial v_1}\frac{\sigma'(v_1)-\sigma'(v_2)}{\sigma(v_1)-\sigma(v_2)}$, is negative when $v_1 < v_2$, that is, $\frac{\sigma'(v_1)-\sigma'(v_2)}{\sigma(v_1)-\sigma(v_2)}$ is decreasing in $v_1$ with fixed $v_2$ when $v_1 < v_2$; monotonicity in $v_2$ with fixed $v_1$ can be proved by the same discussion.

**Proof of (iii).** Regarding the MT and ST labelings, let $y = h_{\text{thr}}(u; \bar{b})$ under the assumption $\bar{b}_1 \leq \cdots \leq \bar{b}_{K-1}$, which implies that $\bar{b}_1 \leq \cdots \leq \bar{b}_{y-1} \leq u \leq \bar{b}_y \leq \cdots \leq \bar{b}_{K-1}$. Regarding the LB labeling for the likelihood model (6), one has that, with the abbreviations $\sigma_k := \sigma(\bar{b}_k - u)$ for $k = 1, \ldots, K$,

$$\begin{aligned}
L_j(u) &= \sum_{k=1}^{K} \{\sigma_k - \sigma_{k-1}\}|j - k|, \\
&= |j - 1|\{\sigma_1 - \sigma_0\} + |j - 2|\{\sigma_2 - \sigma_1\} + \cdots + 2\{\sigma_{j-2} - \sigma_{j-3}\} + \{\sigma_{j-1} - \sigma_{j-2}\} \\
&\quad + \{\sigma_{j+1} - \sigma_j\} + 2\{\sigma_{j+2} - \sigma_{j+1}\} + \cdots + |j - K + 1|\{\sigma_{K-1} - \sigma_{K-2}\} + |j - K|\{\sigma_K - \sigma_{K-1}\} \\
&= -|j - 1|\underbrace{\sigma_0}_{0} + \left\{\sum_{k=1}^{j-1} \sigma_k\right\} - \left\{\sum_{k=j}^{K-1} \sigma_k\right\} + |j - K|\underbrace{\sigma_K}_{1} \\
&= \sum_{k=1}^{j-1} \sigma(\bar{b}_k - u) + \sum_{k=j}^{K-1} \{1 - \sigma(\bar{b}_k - u)\},
\end{aligned} \tag{35}$$

for every $j \in [K]$. Simple calculations show that $\sigma(\bar{b}_k - u) \leq 0.5$ for $k = 1, \ldots, y - 1$ and $\{1 - \sigma(\bar{b}_k - u)\} \leq 0.5$ for $k = y, \ldots, K - 1$, from $\bar{b}_1 \leq \cdots \leq \bar{b}_{y-1} \leq u \leq \bar{b}_y \leq \cdots \leq \bar{b}_{K-1}$ and the assumption on the shape of $\sigma$. One would see that objective function (35) is minimized at $j = y$ because some summands are replaced by ones of 0.5 or more if $j$ deviates from $y$, which concludes the proof. $\square$

The following is an auxiliary lemma for the above-described proof of Theorem 2.

**Lemma 1.** *Suppose that $\sigma$ is non-decreasing and satisfies $\sigma(-\infty) = 0$ and $\sigma(+\infty) = 1$. Define $P(u; a, b) := \sigma(b - u) - \sigma(a - u)$ for $a < b$. Then, one has that*

(i) $P(u; a, b)$ with fixed $a$ and $b$ is symmetric in $u$ around $u = \frac{a+b}{2}$, if $\sigma(-\cdot) = 1 - \sigma(\cdot)$, or if $\sigma$ is differentiable and $\sigma'$ is even.

(ii) $P(u; a, b)$ with fixed $a$ and $b$ is maximized with respect to $u$ at $u = \frac{a+b}{2}$, non-decreasing in $u$ for $u < \frac{a+b}{2}$, and non-increasing in $u$ for $u > \frac{a+b}{2}$, if $\sigma$ is differentiable and $\sigma'(u)$ is even and non-increasing in $u$ if $u > 0$.

(iii) $P(u; a, b)$ with fixed $u$ and $\frac{a+b}{2}$ is increasing with respect to $(b - a)$.

*Proof of Lemma 1.* **Proof of (i).** The assumptions that $\sigma(-\infty) = 0$, $\sigma(+\infty) = 1$, and $\sigma'$ is even imply that $\sigma(-\cdot) = 1 - \sigma(\cdot)$. On the basis of this result, one then has that

$$
\begin{aligned}
P\left(u + \frac{a+b}{2}; a, b\right) &= \sigma\left(b - \left\{u + \frac{a+b}{2}\right\}\right) - \sigma\left(a - \left\{u + \frac{a+b}{2}\right\}\right) \\
&= \sigma\left(\frac{b-a}{2} - u\right) - \sigma\left(-\frac{b-a}{2} - u\right) \\
&= \sigma\left(\frac{b-a}{2} - u\right) - 1 + \sigma\left(\frac{b-a}{2} + u\right),
\end{aligned}
\tag{36}
$$

which implies that

$$
P\left(u + \frac{a+b}{2}; a, b\right) = P\left(-u + \frac{a+b}{2}; a, b\right).
\tag{37}
$$

**Proof of (ii).** The above equation (36) shows that

$$
\frac{\partial}{\partial u} P\left(u + \frac{a+b}{2}; a, b\right) = \sigma'\left(\frac{b-a}{2} + u\right) - \sigma'\left(\frac{b-a}{2} - u\right) = 0, \quad \text{at } u = 0.
\tag{38}
$$

Also, one can show that

$$
\begin{aligned}
\frac{\partial}{\partial u} P\left(u + \frac{a+b}{2}; a, b\right) &= \sigma'\left(\frac{b-a}{2} + u\right) - \sigma'\left(\frac{b-a}{2} - u\right) \\
&= \begin{cases} \sigma'\left(\left|\frac{b-a}{2} + u\right|\right) - \sigma'\left(\frac{b-a}{2} - u\right) \geq 0, & \text{for } u < 0, \\ \sigma'\left(\frac{b-a}{2} + u\right) - \sigma'\left(\left|\frac{b-a}{2} - u\right|\right) \leq 0, & \text{for } u > 0. \end{cases}
\end{aligned}
\tag{39}
$$

Here, for $u < 0$, we used the fact that $\sigma'$ is even, which implies that $\sigma'(\frac{b-a}{2} + u) = \sigma'(|\frac{b-a}{2} + u|)$, and $\sigma'(v)$ is non-increasing in $v$ for $v > 0$ and $\frac{b-a}{2} - u > |\frac{b-a}{2} + u| > 0$; for $u > 0$, we used the fact that $\sigma'$ is even, which implies that $\sigma'(\frac{b-a}{2} - u) = \sigma'(|\frac{b-a}{2} - u|)$, and $\sigma'(v)$ is non-increasing in $v$ for $v > 0$ and $\frac{b-a}{2} + u > |\frac{b-a}{2} - u| > 0$.

**Proof of (iii).** With change of variables $t = \frac{b-a}{2}, v = \frac{a+b}{2}$, we introduce a function

$$
Q(t; u, v) = P(u; v - t, v + t) = \sigma(v - u + t) - \sigma(v - u - t).
\tag{40}
$$

For this function, one has that

$$
\frac{\partial}{\partial t} Q(t; u, v) = \sigma'(v - u + t) + \sigma'(v - u - t) \geq 0,
\tag{41}
$$

since $\sigma$ is non-decreasing (i.e., $\sigma'(u) \geq 0$ for any $u$). $\qquad\square$

Next, we give a proof of Corollary 1.

*Proof of Corollary 1.* If $k < l$, the convexity shows that

$$\ell(k,j) \leq \frac{\{k-j\} - \{k-(j+1)\}}{\{l-j\} - \{k-(j+1)\}} \ell(k,j+1) + \frac{\{l-j\} - \{k-j\}}{\{l-j\} - \{k-(j+1)\}} \ell(l,j)$$

$$= \frac{1}{l-k+1} \ell(k,j+1) + \frac{l-k}{l-k+1} \ell(l,j), \tag{42}$$

and that

$$\ell(l,j+1) \leq \frac{\{l-(j+1)\} - \{k-(j+1)\}}{\{l-j\} - \{k-(j+1)\}} \ell(k,j+1) + \frac{\{l-j\} - \{l-(j+1)\}}{\{l-j\} - \{k-(j+1)\}} \ell(l,j)$$

$$= \frac{l-k}{l-k+1} \ell(k,j+1) + \frac{1}{l-k+1} \ell(l,j). \tag{43}$$

These inequalities imply that $\ell_{k,l}$ is non-positive:

$$\ell_{k,l}(j)$$
$$= \{\ell(k,j) + \ell(l,j+1)\} - \{\ell(k,j+1) + \ell(l,j)\}$$
$$= \{\ell(k,j) + \ell(l,j+1)\} - \left[\left\{\frac{1}{l-k+1} \ell(k,j+1) + \frac{l-k}{l-k+1} \ell(l,j)\right\} + \left\{\frac{l-k}{l-k+1} \ell(k,j+1) + \frac{1}{l-k+1} \ell(l,j)\right\}\right]$$
$$= \left[\ell(k,j) - \left\{\frac{1}{l-k+1} \ell(k,j+1) + \frac{l-k}{l-k+1} \ell(l,j)\right\}\right] + \left[\ell(l,j+1) - \left\{\frac{l-k}{l-k+1} \ell(k,j+1) + \frac{1}{l-k+1} \ell(l,j)\right\}\right]$$
$$\leq 0. \tag{44}$$

Similarly, one can show that $\ell_{k,l}$ is non-negative if $k > l$. □

McCullagh (1980, Section 6.1) has proposed the heteroscedastic extension of (6),

$$\hat{\Pr}_2(y|\boldsymbol{x}; g, \boldsymbol{b}, s) := \sigma\left(\frac{b_y - g(\boldsymbol{x})}{s(\boldsymbol{x})}\right) - \sigma\left(\frac{b_{y-1} - g(\boldsymbol{x})}{s(\boldsymbol{x})}\right) \tag{45}$$

with the scale model $s : \mathbb{R}^d \to (0, \infty)$, and statistical OR studies (Thompson Jr, 1977; Fienberg & Mason, 1979) and (Agresti, 2010, Section 4.2) have also considered another model

$$\hat{\Pr}_3(y|\boldsymbol{x}; g, \boldsymbol{b}) := \sigma(b_y - g(\boldsymbol{x})) \prod_{k=1}^{y-1} \{1 - \sigma(b_{k-1} - g(\boldsymbol{x}))\}. \tag{46}$$

We obtain the following theorem that is similar to Theorem 2 and suggests the efficiency of the IOT labeling for these other models:

**Theorem 4.** *Suppose that $\sigma$ is non-decreasing and satisfies $\sigma(-\infty) = 0$ and $\sigma(+\infty) = 1$ and that $\bar{g} : \mathbb{R}^d \to \mathbb{R}$, $\bar{b}_1 \leq \cdots \leq \bar{b}_{K-1}$, and $\bar{s} : \mathbb{R}^d \to (0, \infty)$.*

*(i)* $\arg\min_{j \in [K]} \sum_{k=1}^{K} \hat{\Pr}_2(k|\boldsymbol{x}; \bar{g}, \bar{\boldsymbol{b}}, \bar{s}) \ell(j,k) = h_{\text{thr}}(\bar{g}(\boldsymbol{x}); \bar{\boldsymbol{b}})$ *if $\ell = \ell_{\text{ad}}$ and $\sigma(0) = 0.5$.*

*(ii)* $\arg\min_{j \in [K]} \sum_{k=1}^{K} \hat{\Pr}_3(k|\boldsymbol{x}; \bar{g}, \bar{\boldsymbol{b}}) \ell(j,k) = h_{\text{thr}}(\bar{g}(\boldsymbol{x}); \boldsymbol{t})$ *for some $\boldsymbol{t} \in \mathbb{R}^{K-1}$ if $\ell = \ell_{\text{ad}}$.*

*Proof of Theorem 4.* **Proof of (i).** The statement (i) of Theorem 4 is trivial from the statement (iii) of Theorem 2.

**Proof of (ii).** Regarding the LB labeling for the likelihood model (46), one has that, with the abbreviations $\dot{\sigma}_k := 1 - \sigma(\bar{b}_k - \bar{g}(\boldsymbol{x}))$ for $k = 1, \dots, K$,

$$
\begin{aligned}
L_j(\bar{g}(\boldsymbol{x})) &:= \sum_{k=1}^{K} \hat{\Pr}_3(k|\boldsymbol{x}; \bar{g}, \bar{\boldsymbol{b}}) \ell_{\mathrm{ad}}(j, k) \\
&= \sum_{k=1}^{K} \left( (1 - \dot{\sigma}_k) \prod_{l=1}^{k-1} \dot{\sigma}_{l-1} \right) |j - k|, \\
&= |j - 1|(1 - \dot{\sigma}_1) + |j - 2|\dot{\sigma}_1(1 - \dot{\sigma}_2) + \cdots + \dot{\sigma}_1 \cdots \dot{\sigma}_{j-2}(1 - \dot{\sigma}_{j-1}) \\
&\quad + \dot{\sigma}_1 \cdots \dot{\sigma}_j(1 - \dot{\sigma}_{j+1}) + \cdots + |j - K + 1|\dot{\sigma}_1 \cdots \dot{\sigma}_{K-2}(1 - \dot{\sigma}_{K-1}) + |j - K|\dot{\sigma}_1 \cdots \dot{\sigma}_{K-1} \\
&= (j - 1) - \left( \sum_{k=1}^{j-1} \prod_{l=1}^{k} \{1 - \sigma(\bar{b}_l - \bar{g}(\boldsymbol{x}))\} \right) + \left( \sum_{k=j}^{K-1} \prod_{l=1}^{k} \{1 - \sigma(\bar{b}_l - \bar{g}(\boldsymbol{x}))\} \right),
\end{aligned}
\tag{47}
$$

for every $j \in [K]$. One has that

$$
L_{j+1}(\bar{g}(\boldsymbol{x})) - L_j(\bar{g}(\boldsymbol{x})) = 1 - 2 \prod_{l=1}^{j} \{1 - \sigma(\bar{b}_l - \bar{g}(\boldsymbol{x}))\},
\tag{48}
$$

is non-decreasing in $j$ with fixed $\bar{g}(\boldsymbol{x})$. Therefore, $\arg\min_{j \in [K]} \sum_{k=1}^{K} \hat{\Pr}_3(k|\boldsymbol{x}; \bar{g}, \bar{\boldsymbol{b}}) \ell_{\mathrm{ad}}(j, k)$ is the first index $l$ such that $L_{l+1}(\bar{g}(\boldsymbol{x})) - L_l(\bar{g}(\boldsymbol{x})) \leq 0$, or $K$ if $L_{l+1}(\bar{g}(\boldsymbol{x})) - L_l(\bar{g}(\boldsymbol{x})) > 0$ for all $l = 1, \dots, K-1$. Also, $L_{l+1}(\bar{g}(\boldsymbol{x})) - L_l(\bar{g}(\boldsymbol{x}))$ is non-increasing in $\bar{g}(\boldsymbol{x})$, for each $l = 1, \dots, K-1$. These facts show that $\arg\min_{j \in [K]} \sum_{k=1}^{K} \hat{\Pr}_3(k|\boldsymbol{x}; \bar{g}, \bar{\boldsymbol{b}}) \ell_{\mathrm{ad}}(j, k) = h_{\mathrm{thr}}(\bar{g}(\boldsymbol{x}); \boldsymbol{t})$ with the threshold parameters $t_k$, $k = 1, \dots, K-1$ satisfying $L_{k+1}(t_k) - L_k(t_k) = 0$. □

## C   Proof of Optimality of Independently Optimized Threshold Labeling

We here provide a proof of Theorem 3.

*Proof of Theorem 3.* In this proof, we use the notations $g_i'$, $y_i'$, $R_{k,i}$, and $\bar{\boldsymbol{t}}$ in Algorithm 1. According to the equations (15) and (17), one has that

$$
\begin{aligned}
\frac{1}{n} \sum_{k=1}^{K-1} R_{k,i_k} &= \sum_{k=1}^{K-1} R_k(t_k; \ell, \bar{g}, \mathcal{D}_n) - \sum_{k=1}^{K-1} R_k(-\infty; \ell, \bar{g}, \mathcal{D}_n) \\
&= R(\boldsymbol{t}; \ell, \bar{g}, \mathcal{D}_n) + \sum_{k=2}^{K-1} R_k(+\infty; \ell, \bar{g}, \mathcal{D}_n) - \sum_{k=1}^{K-1} R_k(-\infty; \ell, \bar{g}, \mathcal{D}_n)
\end{aligned}
\tag{49}
$$

for the threshold parameters $\boldsymbol{t}$ whose elements $t_k$, $k = 1, \dots, K-1$ are $t_k = c_{i_k}(\bar{g}, \mathcal{D}_n)$ with the indices $i_1, \dots, i_{K-1}$ such that $1 \leq i_1 \leq \cdots \leq i_{K-1} \leq n + 1$. Here, the first term of the right-hand side of (49) is the empirical task risk for the threshold labeling with the threshold parameters $t_1, \dots, t_{K-1}$, each of which is a midpoint between $g_{i_k-1}'$ and $g_{i_k}'$, and the second and third terms are constant with respect to the indices $i_1, \dots, i_{K-1}$. Thus, minimization of $\sum_{k=1}^{K-1} R_{k,i_k}$ regarding $i_1, \dots, i_{K-1}$ amounts to minimization of the empirical task risk for the threshold labeling regarding $i_1, \dots, i_{K-1}$ as far as the solutions of the former problem keeps the ascending order. The former minimization can be performed with Algorithm 1 (independent minimizations of $R_{k,i_k}$ regarding $i_k$ for $k = 1, \dots, K-1$). Thus, it can be found that, under the assumption of this theorem, the IOT labeling by Algorithm 1 minimizes the empirical task risk among the class of threshold labelings. □

