# OpenReview forum: "Modified Threshold Method for Ordinal Regression"
_TMLR — Rejected by TMLR_

### Review · Reviewer_YTfd · 2022-08-16

**Summary Of Contributions:**

The submission presents a new method for ordinal classification that takes the intermediate 1-dimensional numeric output (called "1DT") of a machine learning model trained using one of the existing surrogate loss functions designed for ordinal classification and computes threshold parameters that can be used to turn this numeric output into a classification. Experiments on three age estimation datasets, with several existing surrogate losses, and using deep learning, show that the thresholds generated by the proposed method yield slightly lower MAE and MSE than existing approaches for setting the thresholds.


**Requested Changes:**

Suggested changes to strengthen the work:

- Please change the threshold on the p-value.

- Please include results for other domains (e.g., some domains used in deep regression).

- Please include results for direct regression.

**Strengths And Weaknesses:**

Strengths:

The paper presents an interesting and quite extensive discussion of the very specific problem of setting thresholds for ordinal classification when given a "1DT".

The results show small but quite consistent improvements across the board. For one surrogate loss (SVOR), the improvements are quite large (but this loss still always performs worse than the other surrogate losses).

Weaknesses:

The experimental results are all based on age estimation using images of faces. Benchmark datasets from other domains would strengthen the findings.

The threshold for the p-values in the experiments is 0.1, which seems too high. It should be changed to 0.05 at least.

The test tasks are based on minimizing absolute and squared error respectively. It would be instructive to include results where direct regression is performed based on these loss functions.

-------

Some minor issues with wording, etc.:

Abstract

"they learn a one-dimensional transformation (1DT) of the explanatory variable" -- "explanatory variable" does not seem the correct term here

"among intervals of the number of classes" -> "among the intervals pertaining to the different classes"?

"no theoretical rationality" -- this seems too strong (the same applies to "no mathematical rationality" elsewhere in the paper.

Introduction

"They typically learn a 1DT of the explanatory variable (and bias parameters of the number of classes minus one)" -- see above

"its underlying statistical model has a strongly restricted representation ability" -- surely, this depends on the complexity of the model representing the conditional probability

"with the computational complexity of quasi-linear order regarding the training sample size" -- what does this mean and where in the paper is it shown?

Section 2.1

Problem with typesetting in the notation for 1D space of real numbers.

Section 2.2

"It should be noted that the significance of this ordering constraint is determined in relation to the data distribution and the surrogate loss function used together and it is not always effective in improving the classification performance." -- it does not seem entirely clear what this means; anyway, what is the evidence for this statement?

"Next, they build" -> "Next, 1DT-based methods build"

Section 3.1

Proposition 2: "especially if" -- this wording does not seem appropriate here

Section 3.4

Is this theorem circular? Unless "almost everywhere" is very loose, this surely depends on the amount of training data

Section 4.1

"forimproving"

Figure 2: what do the colours mean (i.e., where is the legend)?

Algorithm 1

What does "will be easy to hold mean"?

---

> ### Author Response · Authors · 2022-08-21
> **1st Reply to Review by Reviewer YTfd**
>
> We greatly appreciate the constructive review. We here reply pointed Weaknesses and Requested Changes. But, we wait until all 3 reviews have been submitted before submitting a revised PDF, as the recommendation by TMLR.
>
> ---
> Regarding the additional experiments:
> Thank you for a constructive advice.
> Please give me more time to think about the response (we will write the 2nd reply).
>
> ---
> __p-value__:
> We changed the p-value 0.1 to 0.05.
>
> ---
> __direct regression__:
> First, for example, "$\min_f \frac{1}{n}\sum_{i=1}^n|f(x_i)-y_i|$ with discrete-valued prediction $f:\mathbb{R}^d\to[K]$" is not continuous optimization, and difficult to execute computationally.
> Second, "$\min_f \frac{1}{n}\sum_{i=1}^n|f(x_i)-y_i|$ with real-valued prediction $f:\mathbb{R}^d\to\mathbb{R}$" is not the OR problem defined in Sec.2.1.
> We understood that "direct regression" that reviewer suggested is the second one.
> "direct regression" is not the OR problem, and out of our consideration.
> We consider that experiments of "direct regression" are interesting but may confuse the readers.
>
> ---
> __Abstract__:
> We agree that "no theoretical rationality" is too strong.
> We delete or modify this expression, also in other parts.
>
> ---
> __Introduction__:
> For example, multinomial logistic regression uses $K$-dimensional transformation of the explanatory variable, while the OLR (reviewed in 3.4) uses a 1DT.
> Owing to the use of the 1DT, an underlying statistical model of the OLR has a strongly restricted representation ability:
> We add "owing to the use of the 1DT".
>
> The quasi-linear order is $O(n \log n)$ and stems from the sorting operation (see the modified beginning of Sec.4.2 and above Thm.3).
>
> ---
> __Sec.2.1__:
> 1 of $\mathbb{R}^d.^1$ is a footnotemark symbol.
> This was confusing.
> We modified this.
>
> ---
> __Sec.2.2__:
> Regarding the significance of the ordering constraint:
> Readers may feel the ordering constraint of the bias parameters overly useful due to a sense for "order" in ordinal regression.
> We have written the note to caution against this sensory understanding.
> However, this note violated our policy, "we do not discuss the goodness of the learning procedure".
> So we deleted this note.
>
> ---
> __Sec.3.1__:
> We modify "especially if" to "if" in Proposition 2.
>
> ---
> __Sec.3.4__:
> "almost everywhere" was wrong (we wrote it with the intention of "almost surely").
> This statement does not depend on the amount of training data $n$, since the surrogate risk minimizer is defined as the minimizer of $\mathbb{E}[\phi(g(X),b,Y)]$ not of $\frac{1}{n}\sum_{i=1}^n\phi(g(x_i),b,y_i)$.
> We correct this statement to "for any $x\in\mathbb{R}^d$ in the support of the distribution of X".
>
> ---
> __Sec.4.1__:
> We modify "forimproving" to "for improving".
>
> ---
> __Fig.2__:
> We add the legend to the caption of Figure 2.
>
> ---
> __Alg.1__:
> We change "easy" in the sentence "will be easy to hold" to "more possible".
> Alg.1 sequentially determines the threshold parameters $\bar{t}\_k$ for $k=1,\ldots,K-1$ in the ascending order.
> There exist candidates $c\_{i\_k,j}$ for $j=1,\ldots,n_k$ for threshold parameter $\bar{t}\_k$.
> If $\bar{t}\_k$ is set to a larger candidate, it will become less possible to set $\bar{t}\_{k+1}$ from $c\_{i\_{k+1},j}$'s so that $\bar{t}\_k\le\bar{t}\_{k+1}$.
> The sentence "will be more possible to hold" relates on this concern.

---

> ### Author Response · Authors · 2022-09-02
> **2nd Reply to Review by Reviewer YTfd**
>
> __Regarding the additional experiments__:
> We performed additional experiments for various-domain datasets.
> Please see Section 5.

---

### Review · Reviewer_K5i6 · 2022-08-19

**Summary Of Contributions:**

This paper tackles the ordinal regression problem which consists in affecting an input to a certain discrete category. Further, all categories are assumed to be ordered: for instance, the 3 categories "good", "medium" and "bad". The 1-dimensional transformation (1DT) approach relies on two components. First, a mapping $g$ transforming any multidimensional input $x \in \mathbb{R}^d$ to a scalar value. Second component, a labeling function $h$ mapping any real number to a discrete category ranging from 1 to K. A natural and popular labeling approach consists in using threshold to discretize the real line.
Both components ($g$ and $h$) can be learned from an i.i.d. dataset of input-target pairs $(x_i,y_i)$ valued in $\mathbb{R}^d \times [K ]$ so as to minimize an empirical loss $\frac{1}{N}\sum_{i=1}^n loss(h\circ g(x_i), y_i)$.
This paper focuses on the learning of the second component, namely the labeling function $h$, given some 1DT $g$.
After reviewing and providing theoretical guarantees for existing labeling methods (minimum threshold, summation threshold, nearest-neighbor threshold, labeling based on likelihood ), the authors describe their own new method called independently-optimized threshold labeling (IOT in short). IOT relaxes the joint optimization of all thresholds so that to minimize each threshold independently, which is more tractable from computational perspective. Numerical experiments on several datasets (MORPH-2, CACD,AFAD ) illustrate the efficiency of the proposed method.

**Requested Changes:**

- either: 1 or 2 additional sentences about the joint optimization problem (existing methods, challenges, extension of IOT to this setting, ...)
- or: nothing if this is already discussed in the paper but I simply missed it

**Strengths And Weaknesses:**

Pros
- The paper is well written and draws a broad picture of the ordinal regression problem
- related work is well discussed and theoretical guarantees are also given for already existing methods
- a new computationally efficient method called IOT is proposed, relaxing the optimal threshold problem by making independent the optimization over each individual threshold

Cons
- lack of discussion about the full joint optimization problem, over both components $g$ (1DT) and $h$ (labeling function)
- how can IOT extend to this joint optimization framework?

---

> ### Author Response · Authors · 2022-08-20
> **1st Reply to Review by Reviewer K5i6**
>
> We greatly appreciate the constructive review. We here reply pointed Weaknesses and Requested Changes. But, we wait until all 3 reviews have been submitted before submitting a revised PDF, as the recommendation by TMLR.
>
> ---
> __joint optimization problem__:
> Thank you for helpful comments.
> As we described in Section 2.3, this study does not discuss the goodness of the learning procedure.
> However, improvement of the learning procedure (joint optimization problem) is still important theme.
> We write the following in Section 6, "Conclusion and Future Prospect":
>
> We are also interested in the selection of the learning procedure, especially the surrogate loss function, of the threshold method.
> One may be able to take systematic discussion on the goodness of the loss function by fixing components of the threshold method other than the loss function to the optimal ones.
> In such discussion, the OT and IOT labelings will serve as the optimal other components.
> This is a future prospect.

---

### Review · Reviewer_jaFP · 2022-08-29

**Summary Of Contributions:**

The authors consider the setting of ordinal regression/classification (OR) and a group of threshold methods that maps the sample to the score (one-dimensional transformation (1DT)). The mapping is learned using one of the few discussed storage loss functions. In this group of methods, the final class is selected by checking to which interval between thresholds score belongs. The authors analyze a few such methods under a unified notation of threshold functions. They demonstrate that these methods can become sub-optimal for the minimization of task risk. The authors propose a new way of finding the labeling function for the learned mapping function. The proposed method, named independently-optimized threshold (IOT) is based on the simple idea of finding a set of thresholds that, combined with learned mapping, minimize task loss on a provided dataset. The thresholds are selected from a discreet set of values that are defined by intervals of scores of two consecutive samples in the dataset that were sorted according to that scores. However, instead of considering all possible sets of thresholds (computationally costly), the methods select thresholds independently using the relaxed formulation of the problem. The proposed algorithm is evaluated for different surrogate losses and compared with other threshold methods on three datasets in the domain of age estimation from images. The proposed approach improves task loss in some cases, at the same time almost never being worst than other methods.

**Broader Impact Concerns:**

I have no concerns about the ethical implications of this work.

**Requested Changes:**

I believe the clarifications about things mentioned in the weaknesses would greatly improve the accessibility of the paper.

Formatting improvements:
- Figure 2 lacks a legend for colors.
- In TMLR style, descriptions of tables should be under to have correct spacing. In the submitted manuscript, the descriptions are on the top with no spacing between them and the tables.
- The paper is long, and a first-time reader might not remember what MT, ST, NNT, LB, and others mean. Why not use full names, at least in the titles of sections that correspond to these methods?
- I think, The equation (3) doesn't really need operator $(\cdot)^+$, which is only used in this single equation. This reduces readability.
- Losses are identified with names in subscript and threshold functions with names in superscript. It seems to be a bit inconsistent.

Small language mistakes/suggestions I noticed:
- "[...], as long as the obtained threshold parameters t¯ eventually follow the appropriate [...]" -> follow
- "The CACD dataset could be downloaded [...]" -> can?
- "[...] the MT and ST labelings become same" -> the same
- "Note that Cao et al. (2020) declares that their tried method [...]" -> declare

**Strengths And Weaknesses:**

First, please note that I'm not very knowledgeable about ordinal regression methods, so I'm not able to evaluate this paper
from the angle of novelty in regard to the literature in this area, as well as the correctness and completeness of citations.

Strengths:
- The motivation of the paper is good, and the proposed method is elegant and efficient.
- The paper is sound.
- The paper discusses different surrogate losses and threshold-based labeling methods under unified notation.

Weaknesses:
- The paper is difficult to read at the beginning for the less knowledgable readers - as a person with some knowledge about statistical learning theory and classification methods, but not ordinary regression and especially the discussed methods, I find the paper difficult to read and understand. The authors fail to clearly introduce basic concepts behind the methods they discuss, forcing the reader to read the sections multiple times (especially the first three sections) and figure out his/her own intuitions based on the presented equations and checking references to validate their guesses, on the other hand, presenting a lot of less important information and discussing references (what would be a great addition if not the problem with explaining basic intuitions).
- The problems with writing start with the introduction section: Labels/class words are used interchangeably or even together, without explaining, causing confusion. From the start, the author reference Sections, Theorems, Examples, and Algorithms that are a few pages later. The bias parameters are introduced in the second paragraph of the introduction without explaining what their connection to the labeling procedure and numerous listed threshold methods is. These facts are only partially given in the next paragraphs of the introduction. The threshold methods are discussed in the introduction with little information on how they work. I believe the introduction would benefit from a rewrite, with much clear outline of the whole framework at the beginning.
- I think the preliminaries section repeats some of the sins of the introduction sections: Again, the authors introduce bias parameters of some losses without giving intuitions about bias them. Later labeling function is introduced and defined just on g, so the natural question is why we were focusing on bias parameters b so far, and then Table 1 is presented that mixes all these things together. Finally, in Section 3, I guess things may start to make sense (they started for me), where the labeling methods that were discussed are finally properly introduced. After that, the rest of Sections 3 and 4 are a bit easier to understand but are notation-heavy (not sure if that could be improved here).
- The Experimental Section again lacks some basic details: Some basic information about experiments are lacking, e.g., are MAE and RMSE for benchmark datasets defined on real age values of age ranks? I assume on some ranks, as in Cao et al. 2020, if ranks, then how many ranks, and how are they defined? Why are only age-estimation data considered?
- It is hard for me to tell if the results of the experiment comparison really prove the superiority of the proposed method since it's significantly better only on more or less in half of the cases.
- Is $D^n$ the same dataset used to learn $g(x_i)$? In practice (so in experiments), shouldn't the additional, separate dataset be used for finding thresholds with IOT?

---

> ### Author Response · Authors · 2022-09-02
> **1st Reply to Review by Reviewer jaFP**
>
> We greatly appreciate the constructive review. We here reply pointed Weaknesses and Requested Changes.
>
> ---
> __The paper ~ basic intuitions)__:
> We will omit parts that are not strongly necessary in our discussion.
>
> ---
> __The problems ~ beginning__ & __I think ~ here)__:
> We introduce bias parameters of some losses without giving intuitions about bias parameters (and threshold method and 1DT as well).
> We consider that the intuitional interpretation may misleading their mathematical interpretation.
> For example, we consider that the reason why the ST and MT labelings use the learned bias parameters as threshold parameters comes from the (graphical, non-mathematical) intuitional interpretation of the 1DT-based method (and they are not good);
> see for example, a figure "Fixed Margin" in (Shashua & Levin, 2003),
> Because we are discussing the intuitive and subtle notion "order", paradoxically, we believe that our discussion should be based only on mathematical formulas not to make the discussion subtle.
> For example, the 1DT and bias parameters should be understood only on the basis of the surrogate risk minimization.
> We write the paper according to this belief.
> Namely, since the entire paper emphasizes mathematical descriptions and such a description is undesirable to Section 1, Section 1 gives only a short overview, and the subsequent sections give mathematical details.
>
> Also, the MT and ST labelings are convention, and previous works do not mention a natural reason whey they the learned bias parameters as threshold parameters.
> So, we cannot write intuitive relation of the labeling functions in Section 1.
>
> However, we felt that the paper was difficult to read because of unique terminologies.
> In the revised version, we emphasize terminologies by "\textit{\textbf{~}}" (if this violates the style of TMLR, let us know it).
>
> ---
> __The Experimental ~ basic details__ & __It is hard ~ with IOT?__:
> MAE and RMSE are invariant regarding the translation of target and prediction, and ages have no jump.
> Therefore, using either age or rank as the target does not change the results.
> But, we clarify that for these face-age datasets, we treated the age rank as the target variable, in the revision.
>
> The main reason why we used the face-age datasets is to confirm whether the proposed method achieves the performance competitive to the state-of-the-art method in a modern application.
> However, as reviewers' advice, we add the experiments for various-domain datasets in the revision.
> Sorry see again Section 5.
>
> ---
> __"Formatting improvements" and "Small ~ noticed"__:
> Many thanks for the reasonable requested changes.
> But, the style of the tables in the submitted version should be correct.
> We re-confirmed Table 1 of the latex template by TMLR.

---

> ### Author Response · Authors · 2022-09-03
> **2nd Reply to Review by Reviewer jaFP**
>
> This is an additional reply to the comment against the first draft.
>
> ---
> __Is the same ~ IOT?__:
>
> We consider that this idea (data splitting) may look promising, but will not always work.
> For example, if we change 72% "training data for 1DT and bias and threshold parameters", 8% "validation data", 20% "test data" to 60% "training data for 1DT and bias parameters", 12% "training data for threshold parameters", 8% "validation data", 20% "test data", estimation performance for "1DT and bias parameters" will be degraded.
>
> Also, the existing methods (MT, ST, LB labelings) do not adopt such data splitting (they learn 1DT and threshold parameters simultaneously).
> Therefore, we consider that the paper should present the current method (non-data-splitting) as a comparable method.
> Data splitting is an interesting idea, but we think it is a second step (future work).

---

### Decision · Action_Editors · 2022-10-19

**Recommendation:** Reject

**Comment:**

The author considers the one-dimensional transform (1DT) methodology to solve the ordinal regression problem, which projects the data by some scoring function and then quantizes the scores with respect to some loss function of interest. The author then argues that deciding the optimal thresholds for quantization is computationally difficult, and proposes an efficient method based on solving a relaxed version of the quantization task. The resulting IOT algorithm minimizes each threshold independently, which is proved to be optimal in some but not all conditions. The author demonstrates that the proposed method achieves superior performance on several data sets.

The reviewers enjoy the big picture of ordinal regression discussed in this work and appreciate the simplicity of the proposed method. Before the revision, the reviewers' concerns mostly surround experiments and clarity of writing. The authors fixed the writing and formatting significantly and added additional experiments. Although all reviewers are tilting positively on the acceptance of this work after the revision, the reason that the paper cannot be accepted in its current form is that a fundamental claim from the author is not correct. The author claims that solving (13) takes O(n^{K-1}) with brute-force search, and takes this as the motivation for adopting IOT in Section 4.3.

Somehow solving (13), *exactly*, is a typical dynamic programming problem *for any loss*, see https://en.wikipedia.org/wiki/Dynamic_programming . The dynamic programming structure is based on the fact that an optimal solution concerning k thresholds must contain an optimal solution concerning the first k-1 thresholds. So dynamic programming can go from collecting the optimal solutions for one threshold, and using those to compute the optimal solutions for two thresholds, ..., until we can calculate the optimal solutions for K-1 thresholds. The process takes O(n log n) on sorting and O(n K) on dynamic programming. Lin and Li (2006) actually provided an implementation on this:

https://home.work.caltech.edu/~htlin/program/orensemble/

See the function compute_thres_dploss in aggrank.cpp. I thus respectfully disagree with the authors that "the solved threshold parameters generally have no optimality guarantee in minimization of the empirical task risk."

Given that there is an *easy* dynamic programming solution with the same computational complexity, the motivation to do IOT in Section 4.3 is invalid. The author would at least need to quantitatively compare [1] whether IOT is really computationally more efficiency than DP [2] whether the relaxation in IOT causes any performance degradation.

I personally really like the efforts of the author to bring the thresholding models back to attention for ordinal regression. The revisit on post-thresholding method could provide stronger baseline solutions for the community, which shall be appreciated. I admit that Lin and Li (2006) did not conduct a fully study on 1DT possibilities with DP, and this paper provides a nice piece to fill the gap. I understand that the author should not be penalized for not knowing the existence of the implementation. But given that the dynamic programming solution is an *easy* one, the paper cannot falsely claim the necessity for IOT. I hope the author can revise the paper based on carefully studying the DP solution. In that case, I think the paper would end up having a strong impact on the community, and I will then be happy to reconsider the acceptance.

Disclaimer: I try my best to make a decision professionally, without personal bias. Somehow given that I am one of the authors of Lin and Li (2006) and may have blind spots, I am also happy to reconsider my decision if the authors can prove me wrong (i.e. the dynamic programming steps described above or the code cannot exactly solve (13)).

There are two points that I suggest the author to consider in the revision. But given that no other reviewers shouted about those, addressing the two points or not will not affect my future decision on this paper.

(a) Discuss the relationship to Lin and Li (2012). Reduction from cost-sensitive ordinal ranking to weighted binary classification. Neural Computation.

The cost-sensitive ordinal ranking problem discussed in the paper essentially wants to solve (13) in a more general manner, and IMHO 1DT is a special case of the reduction. The connection can help the community understand the big picture better. (Disclaimer: This is my own work and hence I fully respect the author's decision on including it or not.)

(b) Run experiments on standard benchmark data sets provided by Chu and Keerthi (2007). In the "old" days every ordinal regression paper used those data sets for comparison, and it is unclear why the author did not choose to do so.


**Audience:**

Yes!

**Claims And Evidence:**

The main claim that solving (13) takes O(n^{K-1}) and thus needs to design a relaxation method (IOT) is incorrect, see comments to the author.

---

> ### Author Response · Authors · 2022-10-26
> **Thank you for your careful review. We will retry.**
>
> Dear reviewers and AE,
>
> Thank you for your careful review.
>
> We completely missed a dynamic programming (DP) based method in Lin and Li (2006) for searching for threshold parameters that minimize empirical task risk.
> We were informed of that program (aggrank.cpp in https://home.work.caltech.edu/~htlin/program/orensemble/) in the notice of the rejection decision, we studied it, and understood the decision.
>
> An advantage of the IOT labeling that we can think of at this point is that the for-loop in lines 2 to 6 of Algorithm 1 can be accelerated by parallel processing, but we do not find this point very attractive.
> So we will retry TMLR with a revised paper that clarifies the position of our discussion from the discussion of Lin and Li (2006), and that does not propose the IOT labeling, but rather provides another significance (for example, combining DP-based threshold parameter determination methods with 1DT and bias parameter learning methods such as ORBoost-LR, under a modified formulation that distinguishes between bias parameters and threshold parameters).
>
> Thank you very much.

---

> > ### Comment · Action_Editors · 2022-10-27
> > **thank you**
> >
> > Thank you for understanding the decision and for pushing this research direction forward. I look forward to reading a careful study of the possibility of 1DT given the efficient DP procedure in the future.